# Reframing Structure-Based Drug Design Model Evaluation via Metrics Correlated to Practical Needs

**Bowen Gao**[1,2*], **Haichuan Tan**[1,2*], **Yanwen Huang**[3], **Minsi Ren**[4], **Xiao Huang**[5],
**Wei-Ying Ma**[1], **Ya-Qin Zhang**[1], **Yanyan Lan**[1,6†]

[1]Institute for AI Industry Research (AIR), Tsinghua University
[2]Department of Computer Science and Technology, Tsinghua University
[3]Department of Pharmaceutical Science, Peking University
[4]Department of Artificial Intelligence, Westlake University
[5]College of Intelligence and Computing, Tianjin University
[6]Beijing Academy of Artificial Intelligence (BAAI)

## Abstract

Recent advances in structure-based drug design (SBDD) have produced surprising results, with models often generating molecules that achieve better Vina docking scores than actual ligands. However, these results are frequently overly optimistic due to the limitations of docking score accuracy and the challenges of wet-lab validation. While generated molecules may demonstrate high QED (drug-likeness) and SA (synthetic accessibility) scores, they often lack true drug-like properties or synthesizability. To address these limitations, we propose a model-level evaluation framework that emphasizes practical metrics aligned with real-world applications. Inspired by recent findings on the utility of generated molecules in ligand-based virtual screening, our framework evaluates SBDD models by their ability to produce molecules that effectively retrieve active compounds from chemical libraries via similarity-based searches. This approach provides a direct indication of therapeutic potential, bridging the gap between theoretical performance and real-world utility. Our experiments reveal that while SBDD models may excel in theoretical metrics like Vina scores, they often fall short in these practical metrics. By introducing this new evaluation strategy, we aim to enhance the relevance and impact of SBDD models for pharmaceutical research and development. Code and data are available at `https://github.com/bowen-gao/sbdd_practical_evaluation`.

## 1 Introduction

The field of Structure-Based Drug Design (SBDD) has experienced remarkable advancements in recent years, with the development of models such as Pocket2Mol (Peng et al., 2022), TargetDiff (Guan et al., 2023), and MolCRAFT (Qu et al., 2024). These models are at the forefront of enhancing the efficiency and precision of drug discovery by generating molecules designed to effectively bind to specific protein pockets. Despite these technological strides, the practical application of these models in real-world drug development remains a formidable challenge. The crux of this challenge lies in the verification of their efficacy, which is complicated by difficulties in synthesizing and testing these molecules in laboratory settings.

The Vina docking score (Eberhardt et al., 2021; Trott & Olson, 2010) is the **standard metric** used to estimate the binding abilities of molecules generated by SBDD models. It provides an estimate based on an empirical formula, serving as a proxy for binding affinity. Studies have shown that SBDD models can generate molecules with Vina docking scores outperforming reference ligands (Guan et al., 2023; 2024; Qu et al., 2024), suggesting significant potential in the field. However, its reliability

---

*Equal contribution
†Correspondence to `lanyanyan@air.tsinghua.edu.cn`

is increasingly questioned. As shown in Figure 1, Vina scores can be inflated by simply increasing the number of atoms in a molecule or manipulated by various factors, as discussed in Appendix E, revealing a susceptibility to overfitting. This suggests that reliance on these easily manipulated metrics can lead to overly optimistic model evaluations. Additionally, as noted by Gao et al. (2024), recent advances in modeling have improved Vina scores, but estimates of specific binding ability, such as the delta score, remain unchanged or even worse, still falling short of reference ligand performance. Guo et al. (2021) also done experiments showing that some docking algorithms are not even correlated with binding affinity.

Furthermore, the practical synthesis of molecules generated by current SBDD models often proves to be complex and unfeasible, which significantly impedes their validation in wet-lab experiments (Bradshaw et al., 2019; Gao & Coley, 2020). This challenge, compounded by the long-term reliance on flawed metrics such as the Vina docking score, has led to a notable shortcoming in SBDD—**a disconnect from practical applications**. This gap is evident as the outputs of current SBDD models are theoretically promising, but prove challenging to utilize effectively in real-world settings. To

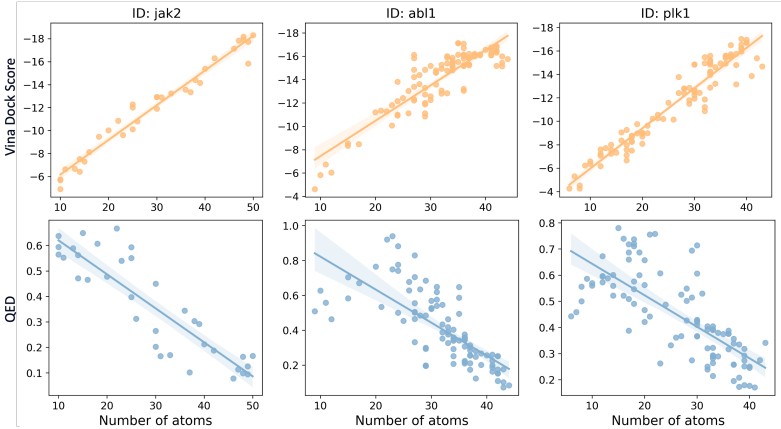

Figure 1: The relationship between Vina Dock Score/QED and number of atoms

address these critical shortcomings in the field of SBDD, we propose a new evaluation framework that extends beyond traditional theoretical estimation-based metrics. Recent attempts to incorporate SBDD-generated molecules into practical drug discovery processes have involved modifying these molecules into existing or more easily synthesizable forms (Moret et al., 2023). Others have used these molecules as reference templates for virtual screening (Shen et al., 2024) or conduct virtual screening using characteristics like scafford of functional groups of the generated molecules (Bo et al., 2024) Inspired by these successful approaches, our proposed evaluation framework includes several new metrics designed to directly assess the practical usability and deployment capabilities of SBDD models from a way that more directly reflects the success rate of wet-lab experiment.

Our framework assesses three levels of evaluation. First, it evaluates the similarity of generated molecules to known active compounds, gauging their potential to be modified into viable drug candidates. Second, it introduces a virtual screening-based metric that directly measures the practical deployment capabilities of these molecules. Third, it continues to consider the estimated binding affinity, albeit in a more nuanced and critically evaluated manner.

By successfully meeting our proposed metrics, an SBDD model is more likely to produce molecules that are not only theoretically effective but also therapeutically viable in real-world drug discovery settings.**Notably, our metrics for using generated molecules in virtual screening directly correlate with and reflect the success rates of wet lab experiments.** This approach aims to bridge the significant gap between theoretical SBDD models and their practical application in the pharmaceutical industry, paving the way for more reliable and efficacious drug development processes.

We conducted extensive experiments using our dataset, which includes data derived from real crystal structures and is divided based on local structural similarities of the pockets, to train and evaluate major SBDD models. Our results show that, from a practical deployment perspective, the molecules generated by current models fall significantly short of matching the quality of reference ligands. Despite achieving high Vina scores, their practical usability metrics reveal a substantial gap. Our

proposed evaluation pipeline is designed to help bridge this gap, offering a direction that could enhance the practical applicability of future SBDD models.

# 2  RELATED WORK

Structure-Based Drug Design involves generating small molecules with potential biological activity for a given protein pocket (Zhang et al., 2023). Several evaluation metrics have been designed to evaluate SBDD models.(Du et al., 2024)Traditionally, biological activity is estimated using AutoDock Vina (Trott & Olson, 2010), a widely used docking software designed to predict the preferred binding orientation of a small molecule (ligand) when bound to a larger protein (receptor) target.

Recently, with the development of deep generative models, several representative models have emerged. These include autoregressive models like AR (Luo et al., 2021) and Pocket2Mol (Peng et al., 2022), diffusion-based models like Targetdiff (Guan et al., 2023), and the newly developed Bayesian Flow Network model, MolCRAFT (Qu et al., 2024).

Typically, these models are trained and tested using the CrossDocked dataset (Francoeur et al., 2020), which is constructed by cross-docking protein-small molecule pairs from the PDBbind dataset. However, this benchmark dataset presents several issues. First, the structures in CrossDocked are generated by docking software instead of real complexes, which may be inaccurate and cannot fully reflect the real interaction pattern. Secondly, the data selection in CrossDocked relies on docking software, which leads to bias in the dataset, as the ligands selected for the training data tend to be favored by the docking software. Third, we observed that many existing models use the test set as a validation set to select checkpoints during training, which risks data leakage. Given these concerns, it is important to adopt a new dataset for a more reliable evaluation of SBDD models.

# 3  METHODS

## 3.1  EVALUATION METRICS THAT REFLECT PRACTICAL NEEDS

Despite recent advancements in deep learning-based generative models for drug design—some even surpassing reference ligands in Vina docking scores—their practical application in pharmaceutical settings remains limited. The ultimate goal is to generate drug-like molecules that specifically bind to intended targets, but current structure-based drug design (SBDD) models fall short in real-world applications.

A significant limitation lies in current evaluation metrics, which fail to accurately assess a model's effectiveness in generating useful molecules. Most of the previous SBDD models published in major machine learning conferences primarily relied on Vina docking scores, biased toward molecules that achieve high scores yet are less effective practically. For example, as shown in Table 1, Vina scores are less predictive than other methods for virtual screening. Additionally, Figure 1 demonstrates that merely increasing molecular size can inflate Vina scores while decreasing Quantitative Estimate of Drug-likeness (QED) values, revealing a vulnerability to overfitting. We also find vina score can be overfitted with more Hydroxyl groups(-OH), less percentage of Nitrogen and Oxygen atoms, and more Halogen atoms. Details can be found at Appedix E. More importantly, despite improvements in Vina docking scores, the delta score—a crucial metric for assessing specific binding capability—still significantly lags behind that of reference ligands. This discrepancy could misdirect model development away from practical needs, underscoring the inadequacy of relying solely on Vina docking scores and highlighting the necessity for more relevant and accurate evaluation criteria.

Another challenge is the synthetic feasibility of molecules generated by deep learning approaches. Although these molecules often achieve high synthetic ability (SA) scores, studies (Bradshaw et al., 2019; Gao & Coley, 2020) indicate they are frequently difficult to synthesize in practice, hindering wet-lab validation and application. This suggests that current theoretical metrics are impractical and do not accurately reflect the ability of models in terms of generating useful molecules.

To address these limitations, we propose reevaluating how model outputs are assessed. Instead of relying on theoretical estimation-based metrics, we advocate for metrics that **directly reflect the usefulness of generated molecules without the need to consider synthetic ability.**

Recent research has involved successfully manually modifying generated molecules into synthesizable structures for wet-lab validation, drawing on the expertise of medicinal chemists who typically adjust molecules based on actives targeting the same biological structures (Moret et al., 2023). Building on this approach, we suggest shifting the evaluation paradigm. Rather than solely aiming for molecules ready for direct wet-lab experiments and binding affinity estimation, we should assess their potential to resemble active compounds. This perspective focuses on the feasibility of transforming generated molecules into practically useful compounds. A similarity or distance-based metric could then be employed to gauge this practical utility, reflecting a more realistic and applicable measure of a generated molecule's value in drug development.

Furthermore, leveraging the capability of generative models in virtual screening to serve as templates for identifying similar compounds has shown promise, achieving significant hit rates (Shen et al., 2024). Thus, we suggest adding a virtual screening metric to evaluate how well a generated molecule can discriminate between active and inactive compounds, providing a direct measure of its utility in drug discovery.

In summary, as shown in Figure 2, our proposed evaluation metrics for assessing the effectiveness of deep learning-based generative models in drug design are structured across three levels:

1. **Similarity To Know Actives:** These metrics evaluate the potential for modification and optimization of generated molecules. They assess how closely these molecules resemble known active compounds, facilitating easier synthesis and optimization.

2. **Virtual Screening Ability:** These metrics determine the ability of generated molecules to distinguish between active and inactive compounds. This is crucial for identifying potential drug candidates that are more likely to succeed in later stages of drug development.

3. **Binding Affinity Estimation:** These metrics theoretically estimate the binding capabilities of generated molecules to target structures. Beyond the conventional vina docking score, we also include other score functions.

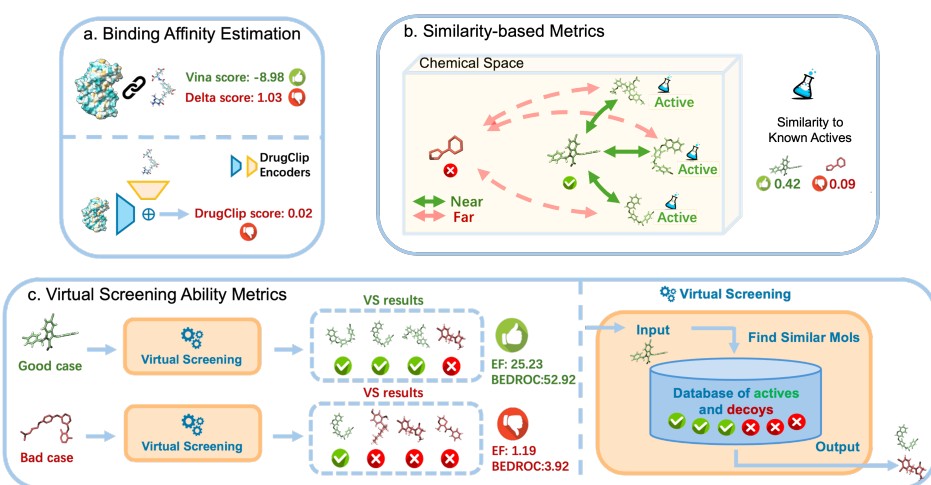

Figure 2: Our three-level evaluation metrics include: (a) Binding affinity estimation, which encompasses the Vina docking score, delta score, and DrugCLIP score; (b) Similarity-based metrics that assess the resemblance between generated molecules and known actives; (c) Virtual screening ability metrics that evaluate the capability of the generated molecules to differentiate between actives and decoys when used as reference templates.

We introduce the details of those metrics.

**Similarity to Known Drugs and Actives**   Our framework maps molecules into a feature space using molecular fingerprints, specifically 2D Extended Connectivity Fingerprints (ECFP) and 3D Extended Three-dimensional Fingerprints (E3FP), which are 1024-dimensional bit vectors. We also use deep learning-based encoders like Uni-Mol (Zhou et al., 2022) and the DrugCLIP (Gao et al., 2023) molecular encoder, which aligns with binding pockets.

Given a target with $N$ known actives $a_1, a_2, \ldots, a_N$, the Active Similarity score is:

$$\text{Active Similarity}(l) = \max_{i \in \{1,2,\ldots,N\}} \left( \sigma(l) \cdot \sigma(a_i) \right).$$

The underlying rationale for the similarity-based metrics is that similar molecules show similar binding behaviour(Boström et al., 2006), and for a generative model to be considered effective, **it must be capable of generating molecules that are at least similar to one of the known active compounds.** Furthermore, to ensure that the generated molecules exhibit "drug-like" properties, they should also demonstrate similarity to one or more known approved drugs.

Table 1: Results from virtual screening using docking software, and using real ligands as templates for similarity searches. Similarity is determined through various molecular fingerprints and deep learning encoders. BEDROC and EF@1 are metrics of vitrual screening. The results demonstrate that using real ligands as templates for virtual screening yields better outcomes compared to virtual screening with the docking software Vina.

| Category | Method | BEDROC ↑ | EF@1 ↑ |
|---|---|---|---|
| Docking | Glide | 40.7 | 16.18 |
| | Vina | - | 7.32 |
| Ligand-based virtual screening | 2D Fingerprints | 39.32 | 24.95 |
| | 3D Fingerprints | 23.77 | 14.30 |
| | Uni-Mol Encoder | 13.39 | 7.48 |
| | DrugCLIP Encoder | **45.43** | **29.23** |

**Virtual Screening Ability**   Relying solely on similarity-based metrics remains insufficient for a comprehensive evaluation. In real-world drug discovery, known active compounds often have decoys—molecules that are structurally similar or share properties but are ineffective against the target. Therefore, if a model predominantly generates decoys that exhibit high similarity to active compounds, it cannot be considered effective. To address this, it is crucial to not only assess the similarity of generated molecules to known actives but also evaluate the model's ability to distinguish between actives and decoys. This ensures the model generates therapeutically relevant molecules rather than misleadingly similar but inactive compounds, providing a more accurate measure of its true potential in drug discovery.

Thus, we introduce our virtual screening-based metrics. Using the molecules generated by our models as references, we retrieve similar compounds from a compound library and evaluate the accuracy of identifying known actives. The library includes experimentally validated actives and decoys—structurally analogous to actives but experimentally confirmed to lack binding affinity. This makes the task of distinguishing actives from decoys particularly challenging. Additionally, we leverage different encoders for virtual screening, offering a flexible approach to better assess model performance across diverse molecular representations.

Specifically, we use the BEDROC and EF metrics to evaluate the effectiveness of virtual screening. BEDROC incorporates exponential weights that assign greater importance to early rankings. In the context of virtual screening, the commonly used variant is $\text{BEDROC}_{85}$, where the top 2% of ranked candidates contribute to 80% of the BEDROC score. The formal definition is shown in equation 1, where $\text{NTB}_\alpha$ is the number of true binders in the top $\alpha\%$. Enrichment Factor (EF) is also a widely used metric, calculated as $\text{EF}_\alpha = \frac{\text{NTB}_\alpha}{\text{NTB}_t \times \alpha}$, where $\text{NTB}_t$ is the total number of binders in the entire screening pool. Table 1 demonstrates that using real ligands as templates for virtual screening yields better results compared to virtual screening with docking software, validating the reliability of our proposed metric.

$$\text{BEDROC}_\alpha = \frac{\sum_{i=1}^{\text{NTB}_t} e^{-\alpha r_i / N}}{R_\alpha \left( \frac{1 - e^{-\alpha}}{e^{\alpha/N} - 1} \right)} \times \frac{R_\alpha \sinh(\alpha/2)}{\cosh(\alpha/2) - \cosh(\alpha/2 - \alpha R_\alpha)} + \frac{1}{1 - e^{\alpha(1 - R_\alpha)}}. \quad (1)$$

The virtual screening capability we use to evaluate SBDD models is central to our proposed metrics. Notably, unlike the Vina docking score, which is a theoretical estimation and shows poor correlation with actual binding affinity, **the enrichment factor from virtual screening is directly correlated**

**with, or even equivalent to, the real hit rate in wet lab experiments.** This means our metric aligns with practical needs and **directly reflects the success rate of SBDD.**

**Binding Affinity Estimation**   Although the focus of this paper is not the theoretical estimation of binding affinity, we still provide more comprehensive metrics to do the estimation. In this benchmark, we continue to use Vina docking scores as one of the metrics to reflect the binding estimation. In addition to the conventional docking score, we also use the delta score proposed by (Gao et al., 2024), which provides a good estimation of the specific binding ability of generated molecules. It provides an unbiased evaluation of whether the generated molecules possess structures that specifically contribute to binding the desired target, rather than structures that overfit the docking software to achieve high docking scores across all targets. To be specifically, for each target $y_i$ and the generated molecules $x_{ij}, j \in 1, 2, ..., m_i$. While the docking score is calculated as $S(x_{ij}, y_i)$, after random sample another target $y_k$, the delta score is calculated as:

$$\textbf{Delta Score}(y_i) = \frac{1}{m_i} \sum_{j=1}^{m_i} (-S(x_{ij}, y_i) + S(x_{ij}, y_k)), \tag{2}$$

for each $i$, we sample $k \in \{1, 2, \ldots, n\}$ with $k \neq i$.

To obtain a more accurate delta score with improved docking scores, we utilized both Glide SP and Glide XP (Friesner et al., 2006) for calculating the delta score. Glide XP is a more precise docking method compared to Glide SP, but it is also more time-consuming.

Beyond docking scores, we also employ a machine learning-based scoring function. DrugCLIP Gao et al. (2023) has demonstrated outstanding performance in virtual screening, making it a valuable evaluation metric for assessing the binding potential of generated molecules.

## 3.2   TEST DATASET

Previous work primarily tested models on the CrossDocked (Francoeur et al., 2020) test set, which has several limitations. It is randomly selected, lacks diversity checks, and is derived from synthesized data rather than real-world crystal structures. Additionally, each pocket in CrossDocked is paired with only one ground truth ligand, even though a single pocket can bind to multiple different ligands.

To address these limitations, we propose a new test set for SBDD models that are created from Mysinger et al. (2012). Our test set comprises 101 targets , with accurately recorded ligand and protein files, and includes a diverse range of protein types as shown in the Appendix I. Each target is supplemented with **a significant number of actives and decoys**, facilitating the evaluation of similarity and distance-based scores as well as virtual screening performance. A robustly generated molecule should effectively distinguish between actives and decoys. On average, each target in our test set contains **224.4 actives and 50 decoys for each active**.The actives and decoys used have been validated through wet-lab experiments. Previous studies have demonstrated that virtual screening methods performing well on these targets can achieve matching results in real-world wet-lab experiments (Wang et al., 2024; Jia et al., 2024), indicating that virtual screening metrics can effectively align with real-world wet-lab performance.

## 3.3   TRAINING AND VALIDATION DATASET

Existing Models are commonly trained and tested on the CrossDocked (Francoeur et al., 2020) dataset, produced by docking software, which consists of synthesized protein-ligand complexes. In contrast, our benchmark utilizes real protein-ligand conformations from experimental crystal structures in PDBbind (Wang et al., 2004). Actually, ligands in PDBbind have a higher docking score as well as delta score for specific binding ability. Details shown in Appendix G

We refined the PDBbind dataset by excluding complexes with nuclear attachment and inaccurately recorded ligands. Then split into a 9:1 training and validation set. To assess the SBDD model's generalization across diverse pockets, we removed samples with similar pockets using FLAPP for pocket alignment and similarity assessment. FLAPP(Sankar et al., 2022) (Fast Local Alignment of Protein Pockets) is a tool used to estimate the structural similarity (alignment rate) between two pockets. We calculated all the FLAPP scores between the training set and test set pockets and removed all pockets from the training set with a FLAPP score greater than 0.6 or 0.9 relative to any

test set pocket. After the removal, the 0.6 version has 12344 complex-ligand pairs remaining while the 0.9 version has 17519 pairs remaining. Details can be found in the Appendix G.

# 4 EXPERIMENTS

## 4.1 TESTED MODELS

We select representative deep learning-based models for structure-based drug design evaluation. For voxel-grid based model, we use LiGAN (Ragoza et al., 2022). For autoregressive models, we choose AR (Luo et al., 2021) and Pocket2Mol (Peng et al., 2022). For diffusion models, we select Targetdiff (Guan et al., 2023). Additionally, MolCRAFT (Qu et al., 2024) is included as a generative model based on the newly developed Bayesian flow network.

## 4.2 RESULTS OF BINDING ABILITY ESTIMATION

Table 2: Evaluation Results for binding ability estimation. Results for Reference Ligands and best results are shown in **bold text**. Atom Efficiency is defined by $Vina\ score/number\ of\ heavy\ atoms$, for reference only.

|  |  | Vina | | Delta Score ↑ | | DrugCLIP score ↑ |
|  |  | Docking Score ↓ | Atom Efficiency | Glide SP | Glide XP |  |
|---|---|---|---|---|---|---|
|  | Reference Ligand | **-9.363** | -0.349 | **2.686** | **3.509** | **0.508** |
| PDBbind 60 | LiGAN | -5.175 | -0.503 | 0.037 | 0.065 | -0.016 |
|  | AR | -7.255 | -0.464 | 0.483 | 0.694 | 0.009 |
|  | Pocket2Mol | -7.640 | -0.479 | 0.531 | 0.553 | -0.005 |
|  | TargetDiff | -9.562 | -0.283 | 0.325 | 0.421 | 0.099 |
|  | MolCRAFT | -9.788 | -0.364 | **0.973** | **1.301** | **0.145** |
| PDBbind 90 | LiGAN | -6.577 | -0.457 | 0.107 | - | -0.000 |
|  | AR | -7.340 | -0.446 | 0.523 | - | 0.007 |
|  | Pocket2Mol | -8.195 | -0.481 | 0.599 | - | -0.006 |
|  | TargetDiff | -9.711 | -0.287 | 0.238 | - | 0.095 |
|  | MolCRAFT | **-9.778** | -0.363 | **1.163** | - | **0.173** |

We first show the metrics that relevant to theoretically binding affinity estimation. In Table 2, it is evident that both TargetDiff and MolCRAFT outperform the reference ligand in terms of average docking scores. However, when considering the Delta Score and DrugCLIP score, they lag significantly behind the reference ligand. Specifically, MolCRAFT, while the best performer among the evaluated methods, still shows a considerable disparity in Delta Score (0.973 vs. 2.686) and DrugCLIP score (0.173 vs. 0.508) when compared to the reference ligand. It is noteworthy that although TargetDiff achieves a competitive docking score, its Delta Score is inferior to those of autoregressive-based methods. This suggests that TargetDiff's high docking score might be the result of overfitting large atom numbers.

## 4.3 RESULTS OF SIMILARITY-BASED METRICS

The evaluation results using similarity-based metrics are presented in Table 3. While MolCRAFT excels in generating molecules that closely resemble known active compounds for specific targets, all models, along with other methods tested, significantly underperform compared to the reference ligand. This suggests that current structure-based drug design (SBDD) models still lack effective conditional generation capabilities, highlighting a key area for further development. The distribution plots for similarity-based metrics across all targets in DUD-E are shown in Appendix F.1 F.2.

## 4.4 RESULTS OF VIRTUAL SCREENING-BASED METIRCS

The results for virtual screening ability metrics are presented in Table 4. MolCRAFT outshines all other methods, regardless of the feature extractor utilized. Nonetheless, it underperforms when benchmarked against reference ligands. Optimistically, using the DrugCLIP molecular encoder for encoding generated molecules for similarity-based virtual screening enables MolCRAFT to achieve a virtual screening efficacy comparable to that of Vina, with an enrichment factor of 5.549 versus Vina's 7.32 (Table 1). This result is particularly encouraging as the speed of virtual screening

Table 3: Evaluation Results for Similarity-Based Metircs on real active molecules. Results for Reference Ligands and best results are shown in **bold text**.

| | | 2D Fingerprints | 3D Fingerprints | Uni-Mol | DrugCLIP |
|---|---|---|---|---|---|
| Reference Ligand | | **0.588** | **0.230** | **0.973** | **0.870** |
| PDBbind 60 | LiGAN | 0.131 | 0.109 | 0.922 | 0.366 |
| | AR | 0.157 | 0.122 | 0.949 | 0.449 |
| | Pocket2Mol | 0.179 | 0.143 | 0.946 | 0.414 |
| | TargetDiff | 0.169 | 0.143 | 0.958 | 0.478 |
| | MolCRAFT | **0.208** | **0.161** | **0.965** | **0.522** |
| PDBbind 90 | LiGAN | 0.136 | 0.112 | 0.931 | 0.397 |
| | AR | 0.161 | 0.125 | 0.951 | 0.468 |
| | Pocket2Mol | 0.185 | 0.146 | 0.945 | 0.419 |
| | TargetDiff | 0.169 | 0.142 | 0.958 | 0.477 |
| | MolCRAFT | **0.214** | **0.163** | **0.966** | **0.547** |

Table 4: Evaluation Results for Virtual Screening-Based Metircs. Results for Reference Ligands and best results are shown in **bold text**.

| | | 2D Fingerprints | | 3D Fingerprints | | Uni-Mol | | DrugCLIP | |
|---|---|---|---|---|---|---|---|---|---|
| | | BEDROC | EF | BEDROC | EF | BEDROC | EF | BEDROC | EF |
| Reference Ligand | | **39.32** | **24.95** | **23.77** | **14.30** | **13.39** | **7.48** | **45.43** | **29.23** |
| PDBbind 60 | LiGAN | 2.306 | 1.079 | 2.216 | 1.007 | 2.024 | 0.837 | 1.581 | 0.655 |
| | AR | 4.938 | 2.567 | 4.407 | 2.196 | 3.266 | 1.501 | 3.698 | 1.796 |
| | Pocket2Mol | 5.976 | 3.054 | 4.192 | 2.047 | 2.335 | 1.033 | 3.667 | 1.827 |
| | TargetDiff | 4.062 | 1.957 | 3.747 | 1.763 | 2.995 | 1.340 | 4.400 | 2.260 |
| | MolCRAFT | **7.584** | **3.953** | **5.521** | **2.792** | **4.868** | **2.357** | **7.265** | **3.968** |
| PDBbind 90 | LiGAN | 2.240 | 1.004 | 2.610 | 1.183 | 2.151 | 0.932 | 1.450 | 0.636 |
| | AR | 4.946 | 2.522 | 4.156 | 2.053 | 3.115 | 1.384 | 4.223 | 2.139 |
| | Pocket2Mol | 6.277 | 3.240 | 4.359 | 2.174 | 2.970 | 1.312 | 3.215 | 1.541 |
| | TargetDiff | 4.431 | 2.174 | 3.888 | 1.861 | 3.967 | 1.828 | 4.122 | 2.117 |
| | MolCRAFT | **9.032** | **4.825** | **6.423** | **3.284** | **5.299** | **2.456** | **9.782** | **5.549** |

with generated molecules is significantly faster than using docking software to dock all candidate compounds. The distribution plots for the enrichment factor of all targets in DUD-E across various models are displayed in Figure 3.

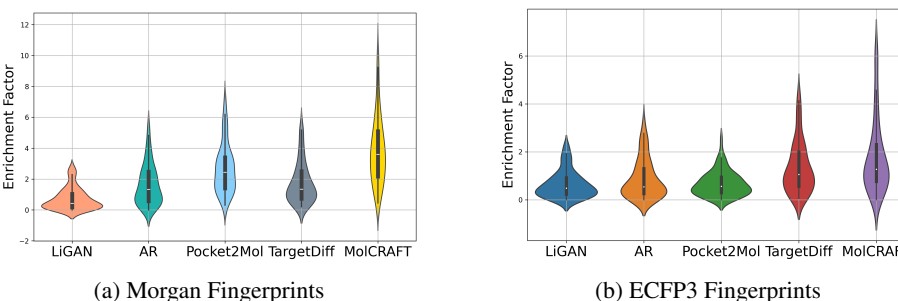

(a) Morgan Fingerprints                    (b) ECFP3 Fingerprints

Figure 3: Distribution plots for Virtual Screening results on all targets in DUD-E with different models.

## 4.5 ANALYSIS

### 4.5.1 INSIGHTS REGARDING PERFORMANCE OF DIFFERENT MODELS

Figure 4 presents radar plots comparing the performance of different models across various metrics in our evaluation. The outer circle represents the metrics of reference ligands. It is evident that

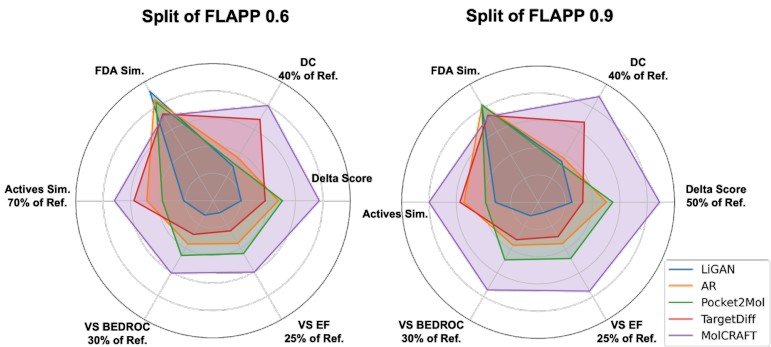

Figure 4: Radar plot shows the performance of different methods on part of our multifaceted metrics. Detailed definition and results refer to FDA-Similarity can be found at Appendix A

MolCRAFT outperforms other methods. Additionally, all methods fall significantly short of the reference ligands, even though some points on the outer circle do not represent the full value of the metrics for reference ligands—such as the virtual screening enrichment factor, which is only 25% of its actual value. Despite this, the models still perform far below the reference ligands.

While achieving impressive Vina docking scores, TargetDiff's overall performance is lacking. The generated molecules exhibit poor delta scores, low similarity to known active compounds, and struggle to distinguish between active and decoy molecules, even when compared to simpler autoregressive methods. This points to a potential flaw in the direction of diffusion-based SBDD models, which, despite their popularity (Guan et al., 2024; Huang et al., 2024), may primarily overfit to Vina scores. The ease of generating large molecules inflates the docking scores but fails to translate into truly useful or relevant molecular structures for drug discovery.

Pocket2Mol ranked second-best for similarity to known actives and virtual screening using fingerprint-based extractors, its performance declines with deep encoder-based methods. The autoregressive nature limits its ability to generate larger molecules. Despite this, its good performance on virtual screening indicates that Pocket2Mol can generate valuable substructures or functional groups, though smaller molecules impact its docking scores negatively. We present some visualizations on Pocket2Mol's ability to generate valuable substructures or functional groups in Appendix D.

MolCRAFT ranks as a top performer across multiple metrics and excels in virtual screening. It achieves results comparable to Vina docking but with greater speed, **demonstrate that using SBDD models to generate molecules, then use those molecules as reference for virtual screening to find potential active molecules is a practical alternative to directly use the generated molecule to do wet-lab experiment.**

Despite achieving higher docking scores, current models still fall short on other proposed metrics such as delta score, similarity to known actives, and virtual screening ability compared to reference ligands. Relying heavily on Vina docking scores can lead to **an overly optimistic view of the effectiveness of SBDD models.** Interestingly, because SBDD models are often judged by their ability to generate molecules with high docking scores, many have shifted focus toward overfitting these scores. For instance, some models use trained Vina predictors to guide sampling, prioritizing the hacking of Vina scores rather than generating reliable and actual effective molecules. This approach can misguide the development of SBDD models. We strongly advocate for evaluating SBDD models using metrics that are **more practically relevant** to drug discovery, rather than solely depending on inaccurate theoretical scoring functions that are easy to be hacked or overfitted.

### 4.5.2 ANALYSIS ON UNBIASED METRICS

In Figure 1, we demonstrate that theoretical estimation-based metrics, such as the Vina score and QED, are prone to overfitting due to their high correlation with molecular properties like atom count. To investigate whether our proposed metrics exhibit similar behavior, we conduct a comparative analysis. As shown in Figure 5, metrics such as delta score, similarity to known actives, and virtual screening performance show no correlation with atom count, indicating their robustness against

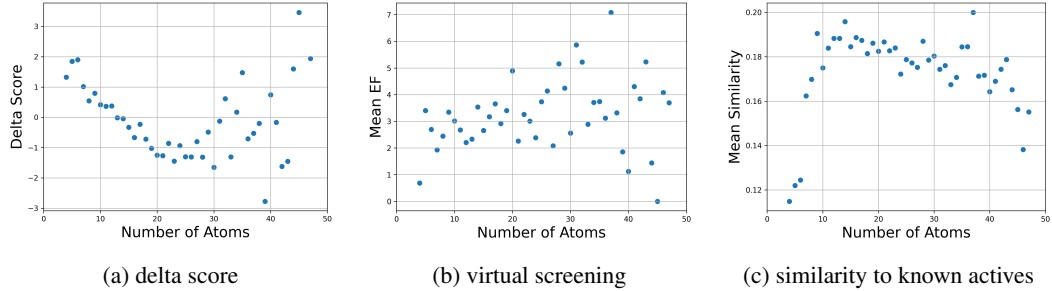

(a) delta score        (b) virtual screening        (c) similarity to known actives

Figure 5: Plots that show the relevance between the atom numbers and different metrics, including (a) delta score (b) virtual screening ability (c) similarity to known actives.

such biases. **Note that as our metrics are based on similarity between molecules, rather than empirical formulas, they are less vulnerable to overfitting or manipulation by specific factors.**

## 4.6 VISUALIZATION OF PROPOSED METRICS

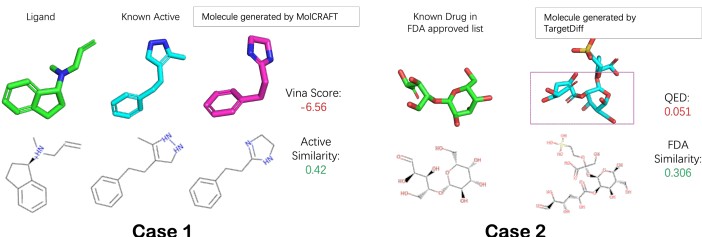

Figure 6: Cases that show the importance of using similarity-based metrics to evaluate the effectiveness of generated molecules

We do some visualizations to show the reason and importance of using our metrics. Figure6(a) shows a molecule generated by MolCRAFT that, although diverging from the reference ligand, aligns with a known active of target AOFB, achieving a high Morgan fingerprint similarity of 0.42 but receiving a bad Vina score of -6.56. This example underscores the necessity of using more practical metrics, rather than solely relying on Vina scores, and emphasizes **the importance of considering all known actives in addition to reference ligands for comprehensive model evaluation.**

In Figure 6(b), a molecule from the TargetDiff model exhibits significant similarity to a molecule on the FDA-approved list, despite its low QED score. This case demonstrates the significance of using similarity as a metric. For details and definition on the FDA-approved list and the definition of similarity, please refer to Appendix A. This suggests its potential as a drug candidate through targeted modifications. This case highlights the importance of our practical related metrics for a more comprehensive assessment of drug-likeness in generated molecules. More visualizations to show the necessity of our proposed metric can be found in Appendix D.

## 4.7 CONCLUSION

In this paper, we present a new benchmark of current Structure-Based Drug Design (SBDD) models with metrics that are related to practical needs. Unlike previous evaluations that primarily relied on Vina docking scores to estimate the efficacy of generated molecules, we advocate for assessing these models from a practical deployment perspective. Our findings reveal that while current methods often outperform reference ligands in terms of docking scores, they fall significantly short when evaluated using the more practical correlated metrics we propose. This discrepancy not only highlights the over-optimism in previous evaluations but also provides valuable insights by reassessing the value of current SBDD models. We hope our evaluation framework will help bridge the gap between generative models and their practical applications.

ACKNOWLEDGMENTS

This work is supported by Beijing Academy of Artificial Intelligence (BAAI).

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

## A FDA-SIMILARITY : AN AUXILIARY SIMILARITY-BASED METRIC

In addition to using similarity to known actives to measure the distance between generated molecules and real actives, we also proposed an FDA similarity metric, comparing the similarity of generated molecules to 2582 FDA-approved drugs to assess the drug-likeness of generated molecules:

$$\text{FDA Similarity}(l) = \max_{j \in \{1,2,...,2582\}} \left( \sigma(l) \cdot \sigma(d_j) \right).$$

where $d_j$ represents the $j$-th molecule in the FDA-approved drug database.

we believe that the 2582 molecules currently approved by the FDA as small-molecule drugs possess certain favorable drug-like chemical properties, such as stability, activity in the human body, toxicity profile, absorption, and proper metabolism. These properties characterize a distribution within a chemical space, and the distance to that chemical space can characterize the possibility a new molecule to be modified to a drug. If a model consistently generates small molecules across various targets that are far from the known drug-like space, the model's ability to generate drug-usable molecules may be questionable.

However, this metric may be overly strict; thus we place it in the appendix and provided the results of various models in Table 5 on this metric for reference.

Table 5: Results for Similarity-Based Metrics including FDA-Approved drug similarity. Results for Reference Ligands and best results are shown in **bold text**.

| | | 2D Fingerprints | | 3D Fingerprints | | Uni-Mol | | DrugCLIP | |
| --- | --- | --- | --- | --- | --- | --- | --- | --- | --- |
| | | FDA | Active | FDA | Active | FDA | Active | FDA | Active |
| Reference Ligand | | **0.348** | **0.588** | **0.139** | **0.230** | **0.975** | **0.973** | **0.749** | **0.870** |
| PDBbind 60 | LiGAN | 0.231 | 0.131 | 0.117 | 0.109 | 0.950 | 0.922 | **0.783** | 0.366 |
| | AR | 0.237 | 0.157 | 0.116 | 0.122 | 0.966 | 0.949 | 0.747 | 0.449 |
| | Pocket2Mol | **0.286** | 0.179 | **0.137** | 0.143 | 0.964 | 0.946 | 0.735 | 0.414 |
| | TargetDiff | 0.209 | 0.169 | 0.132 | 0.143 | 0.971 | 0.958 | 0.683 | 0.478 |
| | MolCRAFT | 0.258 | **0.208** | 0.129 | **0.161** | **0.971** | **0.965** | 0.676 | **0.522** |
| PDBbind 90 | LiGAN | 0.236 | 0.136 | 0.117 | 0.112 | 0.952 | 0.931 | **0.733** | 0.397 |
| | AR | 0.229 | 0.161 | 0.113 | 0.125 | 0.965 | 0.951 | 0.730 | 0.468 |
| | Pocket2Mol | **0.283** | 0.185 | **0.134** | 0.146 | 0.963 | 0.945 | 0.732 | 0.419 |
| | TargetDiff | 0.208 | 0.169 | 0.132 | 0.142 | 0.967 | 0.958 | 0.686 | 0.477 |
| | MolCRAFT | 0.258 | **0.214** | 0.128 | **0.163** | **0.972** | **0.966** | 0.681 | **0.547** |

Pocket2Mol demonstrates the highest similarity to known drugs on the FDA-approved list based on 2D Fingerprints. A notable observation is that the reference ligand shows greater similarity to known active compounds than to FDA-approved drugs in general. In contrast, the generative models display higher similarity to the broader category of FDA-approved drugs rather than to specific known actives.

## B ADDITIONAL BENCHMARK RESULTS FOR OPTIMIZATION-BASED METHOD.

We present the results of RGA (Fu et al., 2022), an optimization-based method, in Table 7. After several rounds of optimization, RGA model can generate molecules with better Vina docking scores. However, the virtual screening metrics are not improved. And the Delta Score is still quite limited.

## C ADDITIONAL BENCHMARK RESULTS ON LIT-PCBA

In addition to the 101 targets used in the main paper, which utilizes decoys designed according to specific rules, we also benchmark the models with 15 additional targets that are derived from LIT-PCBA dataset (Tran-Nguyen et al., 2020), an alternative virtual screening dataset. This dataset consists of decoys that have been actualized in wet-lab experiments and exhibit minimal bioactivity. The threshold used to differentiate between active compounds and decoys in LIT-PCBA is more relaxed, making it a significantly more stringent test.

Table 6: Evaluation Results for optimization-based method RGA.

|  | Binding Affinty | | Virtual Screening | |
|---|---|---|---|---|
|  | Vina docking | Delta score | BEDROC | EF |
| Reference Ligand | -9.363 | 2.686 | 39.32 | 24.95 |
| MolCRAFT | -9.788 | 0.973 | 7.584 | 3.953 |
| RGA Initial | -9.000 | 0.177 | 4.042 | 1.951 |
| RGA Final | -9.665 | 0.286 | 3.888 | 1.850 |

Table 7: Evaluation Results for Similarity-Based Metrics on LIT-PCBA dataset. Results for Reference Ligands and best results are shown in **bold text**.

|  | Similarity to Actives | | Virtual Screening | |
|---|---|---|---|---|
|  | Fingerprints | DrugCLIP | BEDROC | EF |
| Reference Ligand | **0.269** | **0.613** | **4.332** | **3.641** |
| LiGAN | 0.141 | 0.529 | 1.527 | 0.889 |
| AR | 0.157 | **0.558** | 1.879 | 1.361 |
| Pocket2Mol | 0.187 | 0.538 | 2.363 | **1.711** |
| TargetDiff | 0.167 | 0.508 | 2.095 | 1.237 |
| MolCRAFT | **0.189** | 0.556 | **2.498** | 1.577 |

Table 7 shows that LIT-PCBA is indeed a difficult dataset, with an enrichment factor of 3.64 when using actual ligands as a reference, indicating a modest improvement over random selection. MolCRAFT remains the top-performing model overall.

# D MORE VISUALIZATIONS

## D.1 MOLECULE GRAPH

We provide additional visualizations in Figures 7, 8, 9, and 10 to demonstrate the significance of our proposed metrics. Molecules generated by MolCRAFT and Pocket2Mol exhibit good similarity to one of the known active compounds, even though they only achieve moderate docking scores. It is important to note that these molecules are not similar to the original ligands, highlighting the importance of using targets with multiple known actives for evaluating similarity metrics.

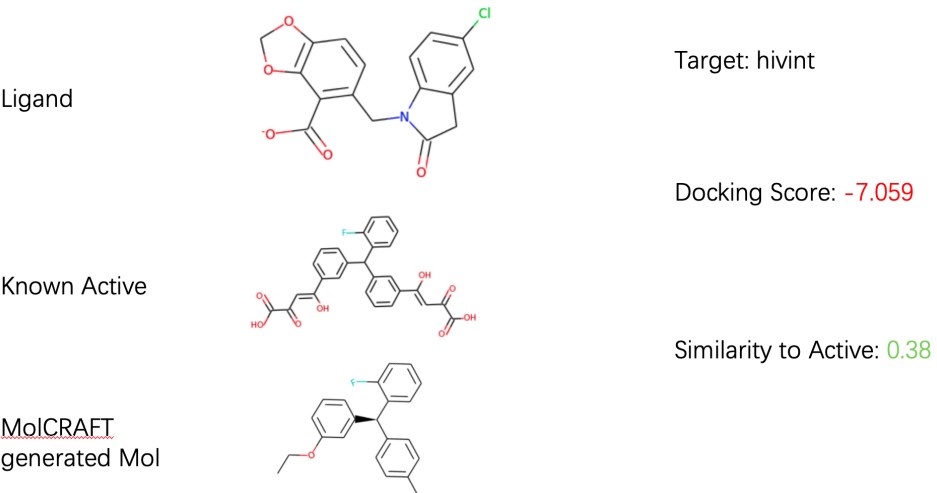

**Figure 7:** Cases that show the importance of using similarity-based metrics to evaluate the effectiveness of generated molecules. Molecule is generated by MolCRAFT.

**Figure 8:** Cases that show the importance of using similarity-based metrics to evaluate the effectiveness of generated molecules. Molecule is generated by MolCRAFT.

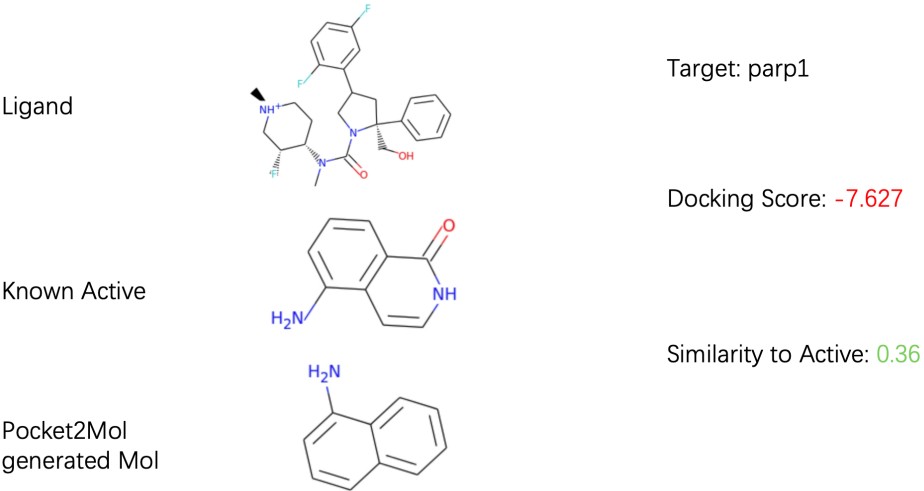

Ligand

Known Active

Pocket2Mol
generated Mol

Target: parp1

Docking Score: -7.627

Similarity to Active: 0.36

Figure 9: Cases that show the importance of using similarity-based metrics to evaluate the effectiveness of generated molecules. Molecule is generated by Pocket2Mol.

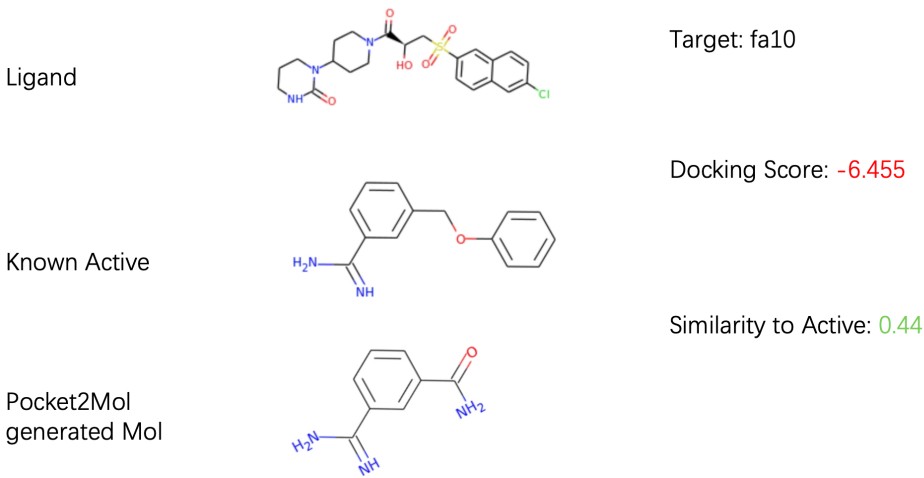

Ligand

Known Active

Pocket2Mol
generated Mol

Target: fa10

Docking Score: -6.455

Similarity to Active: 0.44

Figure 10: Cases that show the importance of using similarity-based metrics to evaluate the effectiveness of generated molecules. Molecule is generated by Pocket2Mol.

## D.2 DOCKING POSES

Here, we present a comparison of the binding poses between real actives and generated molecules with high similarity to actives. Figure 11 and Figure 12 illustrates molecules generated by MolCRAFT, 13 and 14 are generated by Pocket2Mol.

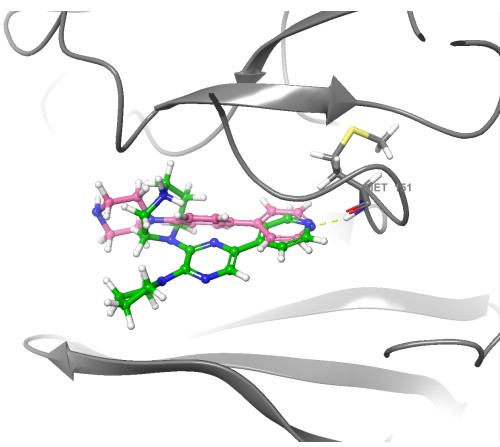

Figure 11: Binding poses between real active molecules (green) and generated molecules (pink) on Target ROCK1. Molecule is generated by MolCRAFT.

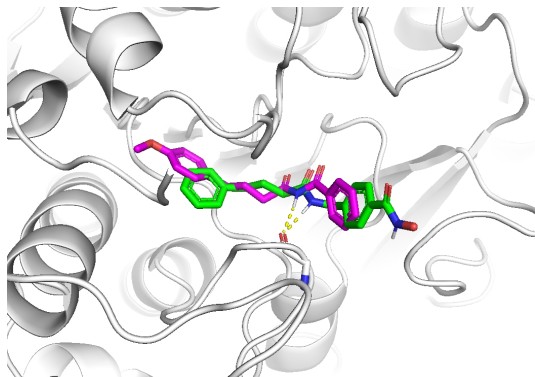

Figure 12: Binding poses between real active molecules (green) and generated molecules (pink) on Target HDAC2. Molecule is generated by MolCRAFT.

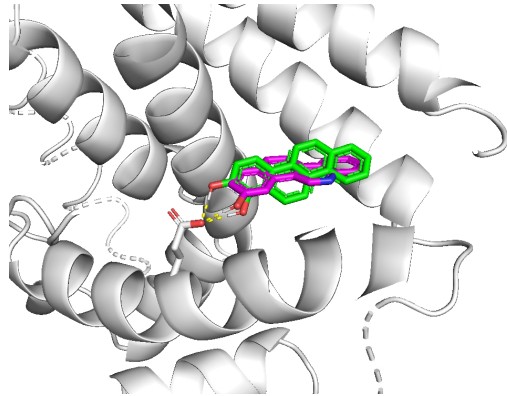

Figure 13: Binding poses between real active molecules (green) and generated molecules (pink) on Target ESR1. Molecule is generated by Pocket2Mol.

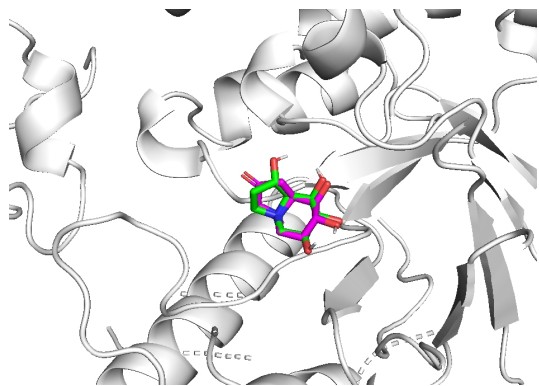

Figure 14: Binding poses between real active molecules (green) and generated molecules (pink) on Target GLCM. Molecule is generated by Pocket2Mol.

# E    ADDITIONAL ANALYSIS ON VINA DOCKING BEING EASY TO BE OVERFITTED

In addition to atom number, we identified several factors that can lead to overfitting or manipulation of Vina docking scores. These factors include the count of hydroxyl groups (-OH), the percentage of nitrogen and oxygen atoms (N+O), and the count of halogens ((fluorine, chlorine, bromine, iodine, and astatine)). The correlations between these factors and Vina docking scores are illustrated in Figures 15, 16, 17.

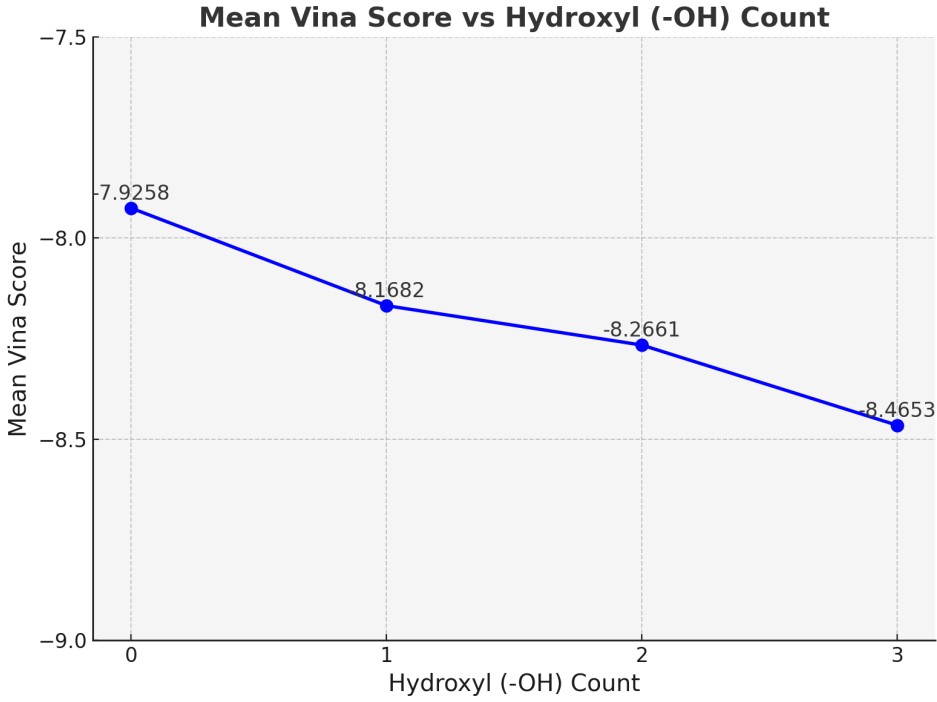

Figure 15: Vina docking scores have a high correlation with the count of Hydroxyl groups.

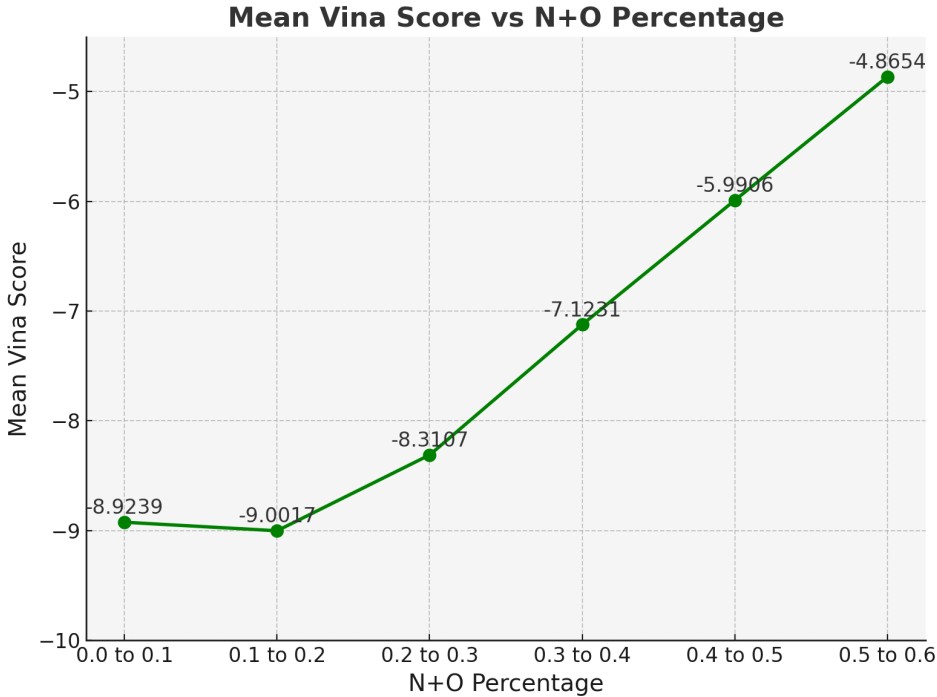

Figure 16: Vina docking scores have a high correlation with the ratio of Nitrogen(N) and Oxygen(O) atoms.

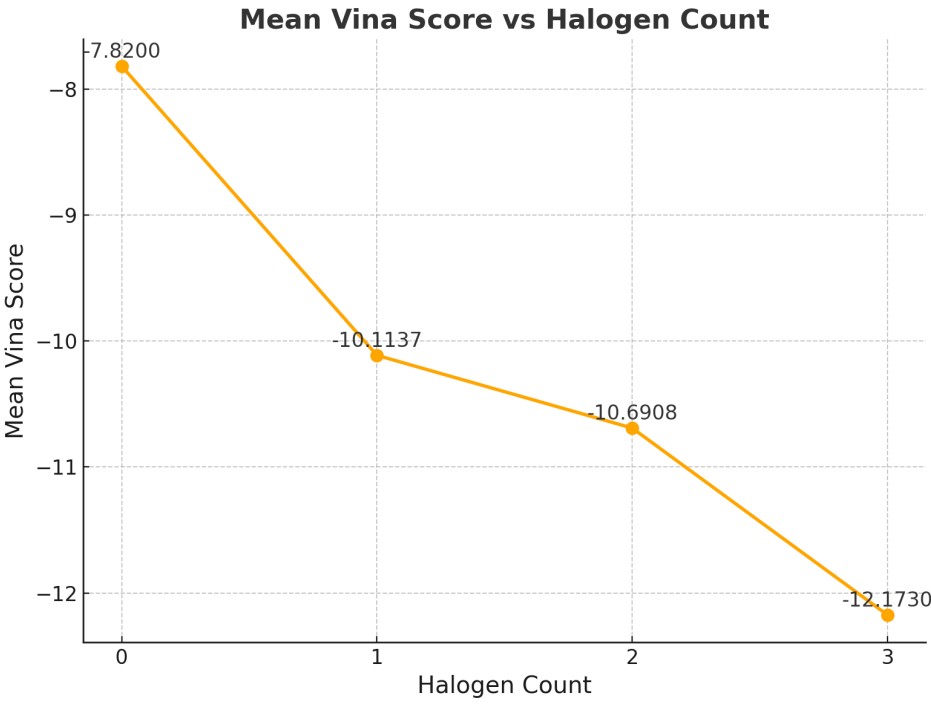

Figure 17: Vina docking scores have a high correlation with the number of Halogen atoms.

# F   ADDITIONAL DISTRIBUTION PLOTS

We provide the violin distribution plots of different targets in the test test for different metrics.

## F.1   DISTRIBUTION PLOT FOR SIMILARITY TO KNOWN DRUGS IN FDA APPROVED LIST

In Figure 18

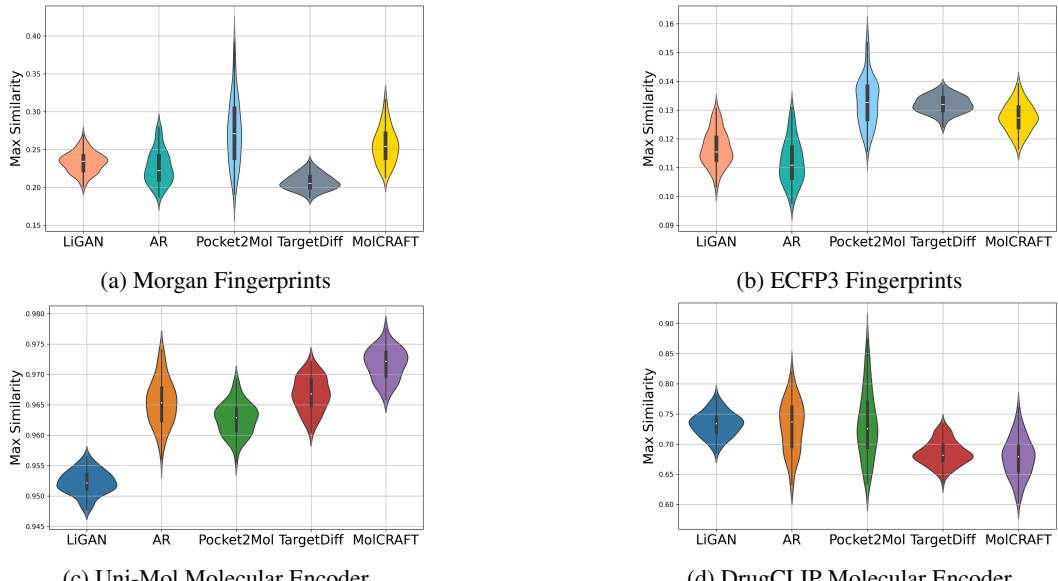

(a) Morgan Fingerprints

(b) ECFP3 Fingerprints

(c) Uni-Mol Molecular Encoder

(d) DrugCLIP Molecular Encoder

Figure 18: Distribution plots for max similarity to all drugs in FDA approves list on all targets in DUD-E with different models.

## F.2   DISTIRBUTION PLOT FOR SIMILARITY TO KNOWN ACTIVES

In Figure 19.

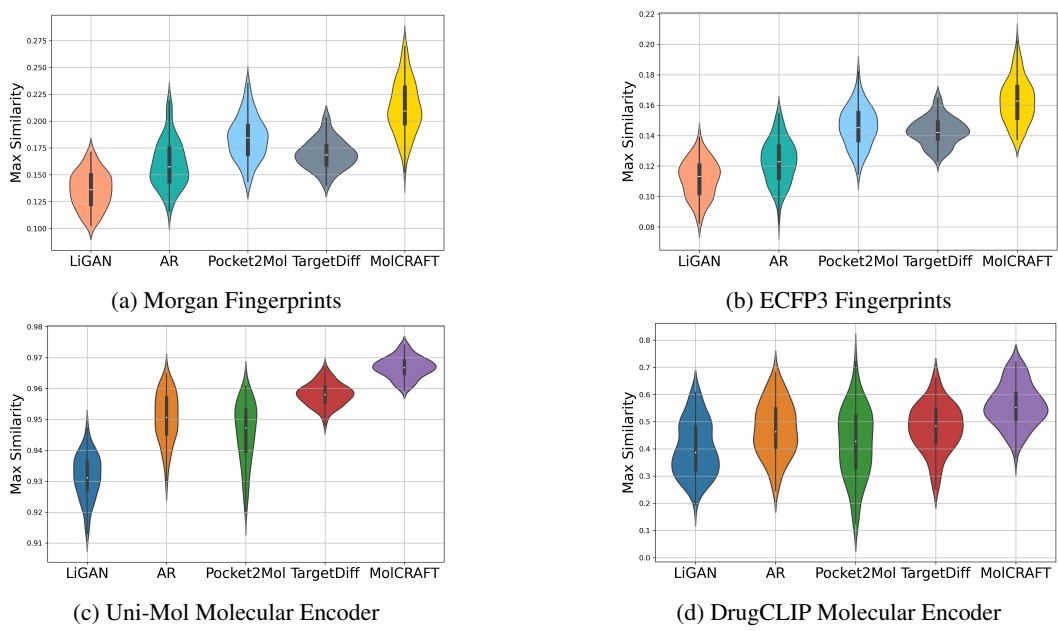

(a) Morgan Fingerprints

(b) ECFP3 Fingerprints

(c) Uni-Mol Molecular Encoder

(d) DrugCLIP Molecular Encoder

Figure 19: Distribution plots for max similarity to known actives on all targets in DUD-E with different models.

## F.3 DISTIRBUTION PLOT FOR VIRTUAL SCREENING

In Figure 3

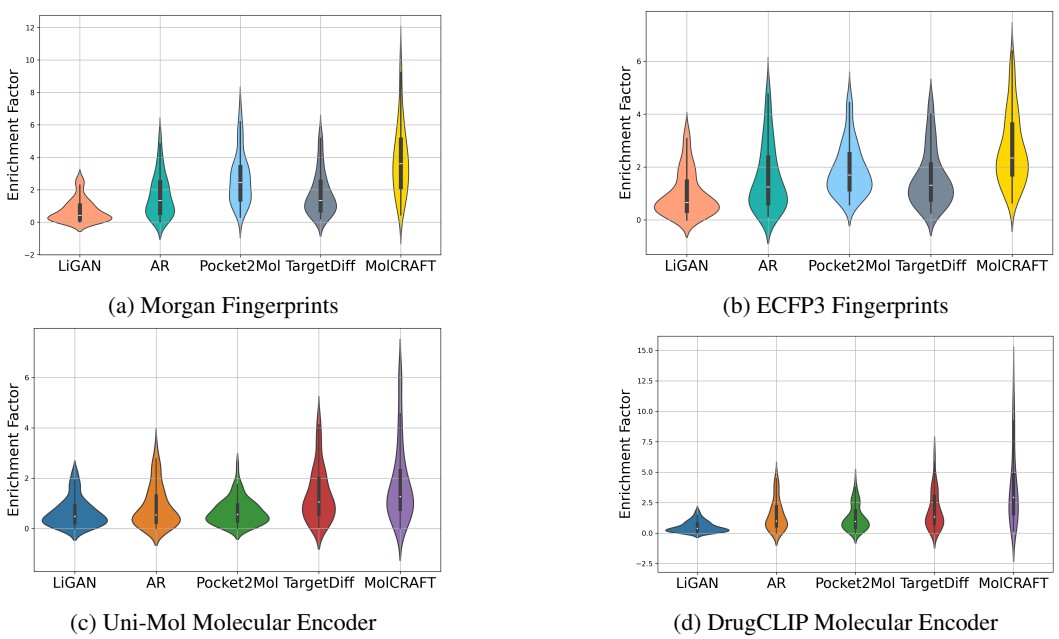

(a) Morgan Fingerprints

(b) ECFP3 Fingerprints

(c) Uni-Mol Molecular Encoder

(d) DrugCLIP Molecular Encoder

Figure 20: Distribution plots for Virtual Screening results on all targets in DUD-E with different models.

## G    COMPARISON OF PDBBIND AND CROSSDOCKED DATASET.

Table 8: Glide Docking score and Delta Score of the pocket with reference ligand on PDBbind and CrossDocked dataset.

|  | Glide Docking | Delta Score |
|---|---|---|
| CrossDocked | -6.37 | 1.05 |
| PDBBind | -7.19 | 1.88 |

## H    TRAINING AND VALIDATION SET DETAILS

We constructed our training and validation sets using data from PDBbind, renowned for its highly reliable, experimentally observed structures. The dataset underwent rigorous filtering to ensure quality and relevance. Initially, we excluded all ligands that RDKit could not correctly interpret, including those with erroneous molecular structures and discontinuous molecules. Subsequently, we selected pockets from target PDB files using a 10 Å distance threshold from the ligand. We further refined the dataset by excluding pockets containing nucleic acids (DNA/RNA) and repairing non-standard residues within the pockets. Rare non-standard residues that could not be repaired were removed. Additionally, protein pockets with fewer than 100 atoms were discarded. This comprehensive filtering process yielded a final set of 19,438 protein pockets, which we then used to construct our training and validation datasets.

To assess the ability of the SBDD generation model to generalize to novel pocket types, we implemented a homology reduction based on pocket structural similarity between the training and test sets. Utilizing FLAPP, we aligned pockets from the training set with those from the test set, quantifying structural similarity through the ratio of successfully aligned amino acids. Figure 21 illustrates the impact of varying FLAPP score thresholds on the number of remaining samples. To strike a balance between removing highly similar pockets and retaining an adequate volume of training data, we selected thresholds of 0.6 and 0.9, resulting in two distinct datasets, as detailed in Table 9. These datasets were subsequently divided into training and validation sets in a 9:1 ratio through random sampling.

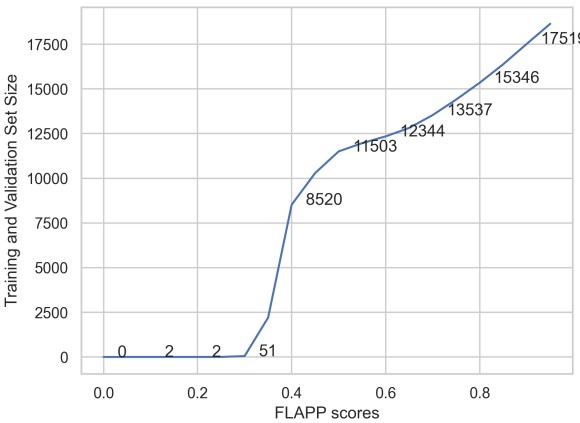

Figure 21: Relationship between FLAPP Score Threshold and Dataset Sizes

Figure 22 illustrates the FLAPP similarity scores between selected targets from the DUD-E dataset and protein pockets from the PDBbind database. The majority of cases have scores clustered in the range of 0.2 to 0.4.

Figure 22: FLAPP score distribution. Each figure is the distribution of FLAPP score between a DUD-E target and all pdbbind pockets.

Table 9: Data sizes for different thresholds

| Threshold (FLAPP Score) | Data Size (PDBbind) |
|:---:|:---:|
| 0.6 | 12,344 |
| 0.9 | 17,519 |

## I  TEST SET DETAILS

We constructed new test sets utilizing data from the well-established virtual screening benchmarks DUD-E and LIT-PCBA. Following the removal of erroneous records, we curated 101 test data points from DUD-E and 15 from LIT-PCBA. These test data cover various categories of protein targets, such as G-Protein Coupled Receptors (GPCRs), kinases, and nuclear receptors. This diversity enables a comprehensive assessment of the model's performance across different protein types. The classification and quantity of these data points are provided in Table 10.

In DUD-E, each target in our test set contains an average of 224.4 active compounds and 50 decoys per active. For LIT-PCBA, each target includes an average of 503.33 active compounds and 176,268.13 decoys. Consistent with our training and validation sets, we defined pockets for SBDD inputs by selecting regions within a 10 Å radius from the reference ligand. This systematic approach ensures a robust and comprehensive evaluation of the model's capabilities across diverse protein-ligand interactions.

Table 10: Distribution of protein target categories in our test set

| Target Categories | DUD-E | LIT-PCBA |
|:---:|:---:|:---:|
| Kinase | 26 | 2 |
| Protease | 15 | 0 |
| Nuclear Receptor | 11 | 4 |
| GPCR | 4 | 2 |
| Miscellaneous | 5 | 0 |
| Ion Channel | 2 | 0 |
| Cytochrome P450 | 2 | 0 |
| Other Enzymes | 36 | 7 |

## J  MODEL TRAINING DETAILS

We employed a new dataset to train and test across five distinct SBDD baselines. All models were trained on a single NVIDIA A100 80GB GPU. The training durations were as follows: MolCRAFT required approximately 30 hours, TargetDiff took around 48 hours, Pocket2Mol also took about 48 hours, AR's main model and frontier model each required 48 hours, and LiGAN training took approximately 30 hours.

## K  MODEL SAMPLING DETAILS

For the 101 + 15 targets in our test set, we sampled 20 small molecules per target using each model. For the autoregressive-based models, AR and TargetDiff, which initially tend to generate smaller molecules, we first sampled 100 molecules and then randomly selected 20 from this set to ensure uniformity in molecule size.

## L  EVALUATION PARAMETERS

For Morgan Fingerprint, we use radius = 2, length of bit vector = 1024.

For E3FP Fingerprint, we use length of bit vector = 1024, radius multiplier = 1.5.

For Uni-Mol molecular encoder, we use the trained weights provided.

For DrugCLIP related models, we use the trained weights provided.

