# OpenReview forum: "Reframing Structure-Based Drug Design Model Evaluation via Metrics Correlated to Practical Needs"
_ICLR.cc/2025/Conference — ICLR 2025 Poster_

### Official Review · Reviewer_8aMH · 2024-10-16

**Soundness:** 3
**Presentation:** 3
**Contribution:** 3
**Rating:** 6
**Confidence:** 4

**Summary:**

The paper provides a comprehensive framework for evaluating structure-based drug design models, using metrics like virtual screening and similarity to known drugs.

The paper is well written. However, the paper has some limitations. Although it acknowledges the difficulty of synthesizing molecules derived from SBDD models, it lacks a systematic approach. A heavy reliance on virtual screening-based metrics can be misleading without experimental validation. For the paper to be more useful and applicable, more real-world case studies should be included.

**Strengths:**

1. very detailed and well-explained in most of the sections.

2.  The propose a more practical framework for evaluating SBDD models, which can be useful.

**Weaknesses:**

1. While the paper acknowledges the difficulty of synthesizing molecules generated by SBDD models, it does not present a systematic approach for addressing this problem.

2. In this paper, virtual screening-based metrics are heavily employed to determine the practical usefulness of generated molecules. While virtual screening is helpful, it could still be misleading if it is not validated with experimental data.

3. This paper needs to include more real-world case studies to be more valuable.

**Questions:**

1. how well does this framework perform when applied to novel targets for which there may be no closely related compounds available?

2. Are there any risks associated with overfitting to the new evaluation metrics?

---

> ### Author Response · Authors · 2024-11-19
>
> Thank you for your valuable review. We have addressed your concerns as outlined below:
>
> ## Regarding synthesizing issue
>
> Thank you for bringing up our discussion on the challenges of synthesizing molecules generated by current SBDD models.
>
> As a benchmark paper, we are not able to provide a solution to improve the synthesizability of current SBDD models. However, from how our evaluation is done, **we have indeed proposed a systematic approach to address this issue**. Our evaluation metrics focus not on directly using the generated molecules in wet-lab experiments but on their potential utility within the drug discovery pipeline. Specifically, we evaluate whether the generated molecules can be easily modified into synthesizable and effective compounds or serve as references for virtual screening to identify existing molecules that are effective against the target.
>
> In our approach, **synthesizability of the generated molecules is not a primary concern because they are not restricted to be directly used in wet-lab experiments.** Instead, the focus is on their **ability to inspire the discovery of practical compounds**. This perspective aligns with several successful virtual screening-based studies that have leveraged molecules generated by SBDD models. For instance, **studies such as TamGen[1] and Pocket Crafter[2] have effectively used generated molecules as references for virtual screening, leading to the identification of viable and effective compounds.**
>
> Thus, our evaluation framework prioritizes the broader utility of the generated molecules in the drug discovery pipeline rather than their immediate synthesizability.
>
>
> [1] Wu, Kehan, et al. "TamGen: drug design with target-aware molecule generation through a chemical language model." Nature Communications 15.1 (2024): 9360.
>
> [2] Shen, Lingling, et al. "Pocket Crafter: a 3D generative modeling based workflow for the rapid generation of hit molecules in drug discovery." Journal of Cheminformatics 16.1 (2024): 33.
>
>
> ## Regarding Whether Virtual Screening-Based Metrics Can Be Misleading
>
> We acknowledge that no evaluation metric can fully replace wet-lab experiments, which remain the definitive test of a molecule’s usefulness. All metrics have inherent limitations and the potential to be misleading. However, our goal is to develop evaluation metrics that are less prone to bias and **more closely aligned with the actual success rates of wet-lab experiments**. Importantly, the virtual screening metrics we propose are evaluated using **compounds validated in real wet-lab settings**, ensuring they **simulate real-world scenarios** rather than relying solely on theoretical estimations. For this reason, we believe our proposed metrics are inherently less biased or misleading.
>
> Furthermore, we fully agree that **relying on a single metric can lead to misleading conclusions**. For example, prior studies and peer-review processes have often **overemphasized the importance of achieving lower (better) docking scores**. This is precisely why we propose our evaluation metrics, which **avoid exclusive reliance on theoretical estimation metrics based on specific factors**. Instead, as detailed in our paper, we evaluate the effectiveness of SBDD models from **three distinct levels**. We introduce the delta score and DrugCLIP score to complement the conventional docking score for theoretical estimation-based metrics, and we also use similarity to known actives as another evaluation level. **By considering these metrics together, we believe the risk of misleading conclusions is minimized**. And for the virtual screening-based metics itself, we employ **various methods**, including 2D fingerprints, 3D fingerprints, and different deep learning-based molecular encoders. We believe that combining these approaches helps mitigate the risk of drawing misleading conclusions.
>
> Overall, while we acknowledge that no metric can be perfect, we are confident that our proposed evaluation framework complements existing evaluation processes and contributes to the better development of SBDD models.

---

> > ### Author Response · Authors · 2024-11-19
> >
> > ## Real world case studies
> >
> > Thank you for suggesting that including more real-world case studies could add value to our work. While conducting actual wet-lab experiments during the rebuttal period is impractical, we would like to highlight existing cases where similar ideas to our framework have successfully led to the identification of useful molecules.
> >
> > Virtual screening-based methods, similar to those we propose, have been employed in several studies and **have successfully used generated molecules from SBDD models as references to identify effective compounds.** For example, the TamGen[1] team utilized commercially available compounds similar to those generated by TamGen. From a 446k commercial compound library, they successfully identified 159 analogs, five of which displayed significant inhibitory effects in wet-lab ClpP1P2 peptidase activity assays. Similarly, Pocket Crafter[2] performed SAR enrichment analysis to extract valuable scaffolds from generated molecules. These scaffolds were then used to perform ligand similarity-based virtual screening. Wet-lab experiments conducted on 2,029 compounds sourced from Novartis’s internal archived diverse library led to the discovery of WM-662 and Compound 1[3], known WBM pocket-binding molecules.
> >
> > These examples demonstrate the practical effectiveness of virtual screening-based approaches in identifying real-world, biologically active molecules. We believe this further supports the utility of our proposed evaluation framework.
> >
> > We are also committed to providing visualization-based case studies to further support our proposed metrics. Some examples are already included in Figure 6, and we have provided additional cases in Appendix Section B for further illustration, including 2D molecule images and 3D docking pose images. Those cases demonstrate the effectiveness and importance of our proposed metrics. The added 3D docking pose images can also be found in the anonymous link: https://anonymous.4open.science/r/tmp-23B1/. It can be observed that for the generated molecules that are similar to known actives, they have similar fragments that generate identical interactions with the target.
> >
> > [1] Wu, Kehan, et al. "TamGen: drug design with target-aware molecule generation through a chemical language model." Nature Communications 15.1 (2024): 9360.
> >
> > [2] Shen, Lingling, et al. "Pocket Crafter: a 3D generative modeling based workflow for the rapid generation of hit molecules in drug discovery." Journal of Cheminformatics 16.1 (2024): 33.
> >
> > [3] Ding, Jian, et al. "Discovery and structure-based design of inhibitors of the WD repeat-containing protein 5 (WDR5)–MYC interaction." Journal of Medicinal Chemistry 66.12 (2023): 8310-8323.
> >
> >
> > ## Regarding the framework applied to novel targets for no closely related compounds available?
> >
> > Firstly, **our framework is for model-level evaluation, and is a benchmark designed to evaluate how well current SBDD models perform using the dataset and evaluation metrics we propose.** For new targets without known actives, evaluating the models with our smilarity-based metrics and virtual screening-based metrics would be analogous to benchmarking a prediction model on a dataset without any labels, which would not provide meaningful insights. **Our framework is not designed to evaluate whether a single molecule can be effective for a new target.**
> >
> > However, our evaluation framework also includes **several metrics that do not require known active compounds.** For instance, the Delta score provides a fair docking score-based evaluation, assessing whether good docking scores result from specific binding or simply overfitting to the docking score itself. Additionally, we incorporate the DrugCLIP score, which demonstrates superior scoring power and accuracy compared to docking software, including Glide.

---

> > > ### Author Response · Authors · 2024-11-19
> > >
> > > ## Regarding risk on overfitting
> > >
> > > Any metric can potentially be overfitted, but we believe our metrics are inherently more resistant to overfitting.
> > >
> > > Unlike theoretical metrics that rely on a scoring function to evaluate molecules, our metrics are based on the similarity between molecules. Scoring functions often include **specific factors that are accessible to users**, making it possible to design algorithms to exploit or overfit these metrics. In contrast, our similarity-based metrics are more robust because it is **impossible to predict which molecules will appear more similar to others** without access to the known actives in the test set. Attempting to do so would effectively involve leaking the test set, which is a clear violation of proper evaluation protocols.
> > >
> > > Therefore, as long as the test set remains inaccessible, it is highly challenging to overfit or hack our evaluation metrics.
> > >
> > > Another source of overfitting arises from biases in the training set. For instance, prior 3D SBDD models have often been trained and evaluated on the CROSSDOCKED dataset. The issue with this dataset is that **it is filtered based on docking software preferences**, meaning the remaining data points tend to align with docking software biases. Models trained on such data can easily overfit to achieve good docking scores without generalizing well. This is why, in this paper, we shift from the CROSSDOCKED dataset to the PDBbind dataset, which contains **real protein-ligand complexes** instead of docked conformations.
> > >
> > > Another notable issue is that previous evaluations of 3D SBDD models often **used the test set as the validation set for model training**, leading to **data leakage and overfitting**. To address this, we provide a standardized data split based on pocket structure similarities, ensuring a more robust and unbiased evaluation process.

---

> ### Author Response · Authors · 2024-11-25
>
> Dear reviewer,
>
> Thank you once again for your thoughtful and constructive feedback. During the rebuttal process, we have incorporated additional real-world case studies that demonstrate how using machine-generated molecules as reference ligands for virtual screening can successfully lead to the identification of hit molecules. This approach also provides a systematic solution to the previously identified challenge that molecules generated by SBDD models are often difficult to synthesize.
>
> Furthermore, we have clarified the nature of our work as a model-level benchmark, emphasizing the extensive efforts we have made to ensure the benchmark minimizes the potential for misleading conclusions. We also provided detailed explanations addressing concerns about the risk of overfitting.
>
> We sincerely hope that our responses have satisfactorily addressed your questions and concerns. If you have any remaining reservations or require further clarification, please let us know. We are more than happy to engage in additional discussions to further refine and strengthen the contribution of our work. Thank you again for your invaluable input.

---

> ### Comment · Reviewer_8aMH · 2024-11-25
> **Thank you for your responses**
>
> Thank you for your thorough responses to my review comments and for addressing the points I raised. I appreciate the detailed clarifications and the effort to incorporate additional points into your framework. After reviewing your rebuttal and revisions, I would like to provide some feedback:
>
> 1. For real-world studies, relying solely on previously published studies raises concerns about the transferability of your work.
> 2. I suggest including a supplementary approach or metric to evaluate framework performance.
>
> Thank you again for your thoughtful responses and for addressing the original feedback. The score remains unchanged.
>
> Best,

---

> ### Author Response · Authors · 2024-11-26
>
> Dear reviewer,
>
> We sincerely appreciate the time and effort you have dedicated to reviewing our paper and rebuttals. We apologize for any lack of clarity in our explanations and would like to provide additional details to address any remaining concerns or questions you may have.
>
> First, We sincerely appreciate your suggestion that "relying solely on previously published studies raises concerns about the transferability of your work." To clarify, the previously published studies we reference are intended to demonstrate real-world cases where using generated molecules as references for virtual screening has successfully led to the discovery of hit molecules. Regarding whether the evaluated models in our paper could lead to the discovery of active molecules in future applications, we believe our evaluation framework and the cases we cited demonstrate this potential. **However, as a machine learning benchmark paper, it is beyond our scope to do actual wet-lab experiments to show that using molecules generated by these models for virtual screening results in finding real active molecules.**
>
> We would like to emphasize that our paper serves as a **model-level benchmark for deep learning-based generative models**. The scope of such work is to test the performance of the models on a fixed test set. Regarding the reviewer’s point about transferability, we understand this refers to the **generalization ability** of machine learning models. We provide explanations from both the computational perspective and the real-world wet-lab validation perspective.
>
> ## Computational perspective
>
> From a computational perspective, in our work, this generalization ability is both tested and ensured with the following methods:
>
> 1. The training set **does not include any** proteins that are structurally similar to those in the virtual screening test set. **We constructed our dataset starting from PDBbind and performed a strict split**. **We remove pockets in the training set that are structurally similar to those in the test set.** Structural similarity was measured using the pocket alignment method FLAPP [1].  We adopted two different FLAPP score thresholds, 0.6 and 0.9, for the splitting process to create two datasets for comparison, aiming to explore the impact of homology on testing performance. Details about this process are provided in Section 3.3 of the paper.
>
> Thus, the targets in the test set are **unseen new proteins for the model**. Our metric is to measure the model’s performance in virtual screening for these new targets. Therefore, if a model performs well on our virtual screening metric, it already demonstrates the model good at conducting virtual screening on previously unseen targets.
>
> 2. The test set we used includes **over 100 targets distributed across 8 categories**, including Kinase, Protease, GPCR, Ion Channel, Nuclear Receptor, Cytochrome P450, Other Enzymes, and Miscellaneous [2]. The diversity of actives calculated by (1- RDK fingerprint similarity) is shown below. We compare the active sets for targets, and the molecules generated by MolCRAFT and TargetDiff for targets. The mean values are shown here:
>
> | Method       | Diversity |
> |--------------|-----------|
> | Active sets  | 0.633     |
> | MolCRAFT     | 0.618     |
> | TargetDiff   | 0.581     |
>
> Results show that the diversity between the active molecules actually has a better diversity than the molecules generated by models. We also provide several distribution plots to demonstrate such diversity:
> - logp: https://anonymous.4open.science/r/tmp-23B1/logp.jpeg
> - molecule weight: https://anonymous.4open.science/r/tmp-23B1/mw.jpeg
> - hydrogen bond acceptors: https://anonymous.4open.science/r/tmp-23B1/hba.jpeg
> - hydrogen bond donors: https://anonymous.4open.science/r/tmp-23B1/hbd.jpeg
>
> **In conclusion, the diversity of target types and the active molecules in the test set provides a reasonable level of confidence in the generalizability of our evaluation.**
>
> Moreover, **this is the standard procedure for evaluating machine learning models across all domains.** We understand the reviewer’s concern that a model performing well on our test set might still fail when applied to a new target. **However, this limitation is inherent to all evaluation metrics and cannot be entirely avoided.** As we have stated, there are no metrics or evaluation methods that can guarantee that a model that performs well on a fixed test set will always perform well on new data.  Even the actual wet lab experiment cannot guarantee it. What we can do is to design a good test set and good split to fairly evaluate the generalization ability of models, and it has been done in our work.

---

> ### Author Response · Authors · 2024-11-26
>
> ## real-world wet-lab validation perspective
>
>
> **The active and inactive molecules in our evaluation framework have been real-world wet-lab validated [2,3]**, ensuring that our evaluation metrics—such as those assessing similarity to known actives and virtual screening capabilities—directly simulate real-world scenarios. This allows us to provide a robust and meaningful benchmark for assessing machine learning models in a way that is both practical and relevant to real-world applications.
>
> We can also provide more evidence on the reliability of using virtual screening-based metrics. **Several studies have demonstrated an alignment between virtual screening metrics and real-world wet-lab performance, even for completely new targets.** For instance, [4] and [5] proposed virtual screening methods that achieved high scores on DUD-E and LIT-PCBA benchmarks while in wet-lab experiments, [4] successfully identified non-covalent inhibitors for the challenging protein target GPX4, which has a flat surface, with an IC50 of 4.17 μM. Meanwhile, [5] identified 12 diverse molecules with significant affinities for NET, a newly solved protein structure. It achieved a hit rate of 15% in the wet-lab experiments which matched the computational metric estimation.
>
> We hope this clarifies our intentions and addresses concerns regarding the transferability and scope of our work. If there are additional steps we should take to further demonstrate the validity of our benchmark, we would greatly appreciate your guidance. **Would the reviewer kindly specify the case studies you would like us to conduct? If there are specific requirements or details for case studies you would like us to address, we would greatly appreciate and happy to provide additional analyses wherever feasible.**
>
> [1] Sankar, Santhosh, Naren Chandran Sakthivel, and Nagasuma Chandra. "Fast local alignment of protein pockets (FLAPP): a system-compiled program for large-scale binding site alignment." Journal of Chemical Information and Modeling 62.19 (2022): 4810-4819.
>
> [2] Mysinger, Michael M., et al. "Directory of useful decoys, enhanced (DUD-E): better ligands and decoys for better benchmarking." Journal of medicinal chemistry 55.14 (2012): 6582-6594.
>
> [3] Tran-Nguyen, Viet-Khoa, Célien Jacquemard, and Didier Rognan. "LIT-PCBA: an unbiased data set for machine learning and virtual screening." Journal of chemical information and modeling 60.9 (2020): 4263-4273.
>
> [4] Wang, Zhen, et al. "Enhancing Challenging Target Screening via Multimodal Protein-Ligand Contrastive Learning." bioRxiv (2024): 2024-08.
>
> [5] Jia, Yinjun, et al. "Deep contrastive learning enables genome-wide virtual screening." bioRxiv (2024): 2024-09.
>
>
>
> # Regarding the evaluation of the whole framework
>
> Regarding the suggestion for a metric to evaluate framework performance, we are unsure of its intended meaning.
>
> If the suggestion implies that we need a metric to assess how effective our evaluation framework is, we believe this is not feasible. As a model-level benchmark, our work focuses on providing insight into which model performs better according to our evaluation metrics. However, the “ground truth” of which model is definitively better is inherently unknown, making such an evaluation impossible.
>
> Alternatively, if the suggestion is to provide a metric that summarizes the overall performance of models across different metrics within our framework, we can address this. For example, we could present a sum of ranks for each model across all metrics or calculate a success rate based on thresholds for various metrics.
>
> Would the reviewer kindly clarify which of our interpretations is correct? We would be happy to provide further explanations.
>
> **Thank you again for the valuable feedback. We would be delighted to further discuss any remaining concerns or questions the reviewer may have.**

---

> ### Author Response · Authors · 2024-11-29
>
> Dear Reviewer,
>
> Thank you once again for your thoughtful review and feedback. We would like to take this opportunity to follow up on our previous response to further clarify and reinforce our position.
>
> Firstly, we would like to reaffirm that our framework is not subject to issues of transferability or generalization, **as it is specifically designed as a benchmark paper**. Our evaluation operates at the model level, rather than assessing the potential of individual molecules for specific protein targets. The targets we selected serve as standardized "exam questions" to which all models must respond, ensuring fairness and validity in our comparisons. Without established knowledge, it is impossible to evaluate newly developed models effectively.
>
> Furthermore, in the context of a benchmark paper, we have made considerable efforts to implement a **rigorously designed data split** and simulations that closely mimic real-world scenarios. **The active molecules used in our study have been validated in wet-lab experiments**, as previously detailed.
>
> As demonstrated in our work and highlighted by another reviewer, **numerous studies and case reports have confirmed that utilizing generated molecules in virtual screening is a reliable and feasible strategy for drug discovery [1, 2, 3, 4, 5]**. Based on this evidence, we firmly believe that **there are no transferability concerns with our evaluation framework**.
>
> To further illustrate the relevance and applicability of our evaluation metrics, we conducted a simulation study on the **relatively unexplored target**, Ras-related protein Rab-2A (UniProt ID: P61019), generating new molecules using the complex structure with PDB ID 1Z0A. In this case study, one of the models evaluated in our framework, MolCRAFT, **successfully generated a molecule that retrieved the only ChEMBL-listed active compound from the ChemDiv molecule library**, which contains 1.64 million compounds. When the generated molecule was used as a reference for virtual screening, the actual active compound ranked among the top 0.15% of all screened molecules, as determined by similarity using Morgan fingerprints. This result highlights the model’s ability to generate biologically relevant molecules, even for targets that have not been extensively explored in prior wet-lab studies. The visualization for the ChemDiv library, the active molecule, and the generated molecule can be found here: **https://anonymous.4open.science/r/tmp-23B1/visualization.jpg**.
>
> Interestingly, when evaluated using docking scores, the generated molecule achieved a moderate-to-low docking score of -6.0. This underscores the comprehensive nature of our evaluation framework, which assesses a model’s performance across multiple facets of the drug discovery pipeline, rather than relying solely on docking performance. This broader approach ensures that generative models are thoroughly evaluated in scenarios that are more directly related to their practical utility in real-world drug discovery applications.
>
> **In summary, the methods and metrics used in our evaluation framework are well-supported by numerous previous studies and can be further validated through research on targets that lack prior experimental studies. As a benchmark, the validity and transferability of this work are ensured by our rigorous dataset splitting process, which enables reliable model evaluation, as well as the use of wet-lab-validated active compounds as labels in our assessment. We hope these explanations address any concerns you may have regarding the robustness of our benchmark. Thank you again for your time and effort in reviewing our paper!**
>
> [1] Wu, Kehan, et al. "TamGen: drug design with target-aware molecule generation through a chemical language model." Nature Communications 15.1 (2024): 9360.
>
> [2] Shen, Lingling, et al. "Pocket Crafter: a 3D generative modeling-based workflow for the rapid generation of hit molecules in drug discovery." Journal of Cheminformatics 16.1 (2024): 33.
>
> [3] Ding, Jian, et al. "Discovery and structure-based design of inhibitors of the WD repeat-containing protein 5 (WDR5)–MYC interaction." Journal of Medicinal Chemistry 66.12 (2023): 8310-8323.
>
> [4] Putin, Evgeny, et al. "Adversarial threshold neural computer for molecular de novo design." Molecular Pharmaceutics 15.10 (2018): 4386-4397.
>
> [5] Bo, Weichen, et al. "Local scaffold diversity-contributed generator for discovering potential NLRP3 inhibitors." Journal of Chemical Information and Modeling 64.3 (2024): 737-748.

---

> > ### Author Response · Authors · 2024-12-02
> >
> > Dear Reviewer,
> >
> > With only 24 hours remaining before the deadline for reviews to post questions, we wanted to follow up to confirm whether our previous responses and case studies have adequately addressed your concerns and clarified any uncertainties regarding our benchmark.
> >
> > We greatly appreciate the time and effort you have invested in reviewing our work. If there are any remaining questions or points requiring further explanation, we would be glad to provide additional details promptly.
> >
> > Thank you once again for your thoughtful review.
> >
> > Best regards,
> >
> > The Authors

---

### Official Review · Reviewer_WYJC · 2024-10-29

**Soundness:** 3
**Presentation:** 3
**Contribution:** 3
**Rating:** 6
**Confidence:** 3

**Summary:**

This paper points out a curial problem in current SBDD models evaluation: The vina docking score metric is easy to inflate and other metrics are hard to test in wet lab. So the paper proposed three-level evaluation metrics: Similarity to Known Drugs and Actives, Virtual Screening Ability and Binding Affinity Estimation. They test on five models and claim that MolCRAFT outperforms other methods

**Strengths:**

1. Provide a novel benchmark to tackle a curial problem in this area
2. Select a diverse set of models on SBDD

**Weaknesses:**

1. Focus on testing deep learning models. Could test more on other based such as RL based or Genetics Algorithm based
2. Table 1's description is too simplified, hard to let people to understand why using real molecules as template is better
3. It would be better to briefly describe some terminology, i.e. FLAPP

**Questions:**

1. The proposed benchmark relied on comparing similarity with known drug molecules, does this affect the search for de novo drug molecules?
2. Could you show the docking pose images of molecules generated by MolCRAFT and Pocket2Mol exhibit good similarity to one of the known active compounds

---

> ### Author Response · Authors · 2024-11-19
>
> We thank the reviewers for their thoughtful feedback. The reviewer recognized that we are addressing a crucial issue in SBDD, which is very encouraging for us. We address reviewer comments below and will incorporate all feedback.
>
>
> ## Q: More non-deep-learning methods should be tested.
>
> We appreciate the reviewer’s valuable feedback.** We have added an evaluation of RGA[1], which is an SBDD method based on reinforcement learning and genetic algorithms.**
>
>
> Table 1: results for RGA for the first round and the final round.
>
> | Method            | Vina Docking | Delta Score | BEDROC  | EF      |
> |--------------------|--------------|-------------|---------|---------|
> | Reference Ligand | -9.363  | 2.686   | 39.32 | 24.95 |
> | MolCRAFT          | -9.788       | 0.973       | 7.584   | 3.953   |
> | RGA Initial       | -9.000       | 0.177       | 4.042   | 1.951   |
> | RGA Final         | -9.665       | 0.286       | 3.888   | 1.850   |
>
> As shown in the table, it is obvious that after several rounds of optimization， RGA model can generate molecules with better Vina docking scores. However, the virtual screening metrics are not improved. And the Delta Score is still quite limited.
>
>
> [1] Fu, Tianfan, et al. "Reinforced genetic algorithm for structure-based drug design." Advances in Neural Information Processing Systems 35 (2022): 12325-12338.
>
> ## Q: A detailed explaination of Table 1.
>
> Thank you for your comments. We have reorganized and provided more detailed explanations for Table 1 in hopes of accurately conveying our ideas. The revised version is as follows:
>
>
> Caption: Results from virtual screening using docking software, and using real ligands as templates for similarity searches. Similarity is determined through various molecular fingerprints and deep learning encoders. BEDROC and EF@1 are metrics of vitrual screening. The results demonstrate that using real ligands as templates for virtual screening yields better outcomes compared to virtual screening with the docking software Vina.
>
> | Category                        | Method            | BEDROC ↑ | EF@1 ↑  |
> |---------------------------------|-------------------|----------|---------|
> | **Docking**                     | Glide             | 40.7     | 16.18   |
> |                                 | Vina              | -        | 7.32    |
> | **Ligand-based virtual screening** | 2D Fingerprints  | 39.32    | 24.95   |
> |                                 | 3D Fingerprints   | 23.77    | 14.30   |
> |                                 | Uni-Mol Encoder   | 13.39    | 7.48    |
> |                                 | DrugCLIP Encoder  | **45.43** | **29.23** |
>
> If you feel this requires further clarification, please let us know. We would be more than willing to provide additional details.
>
> ## Q: A explaination of FLAPP
>
> Thank you for your suggestions. We have further elaborated on flapp and integrated it into the paper. If there are any other terms you believe require further explanation, please feel free to let us know. Here is the description of FLAPP:
>
> FLAPP (Fast Local Alignment of Protein Pockets) [1] is a tool used to estimate the structural similarity (alignment rate) between two pockets. We calculated all the FLAPP scores between the training set and test set pockets and removed all pockets from the training set with a FLAPP score greater than 0.6 or 0.9 relative to any test set pocket. This step was taken to assess the SBDD model’s ability to generalize across structurally diverse pockets.
>
> [1] Sankar, Santhosh, Naren Chandran Sakthivel, and Nagasuma Chandra. "Fast local alignment of protein pockets (FLAPP): a system-compiled program for large-scale binding site alignment." Journal of Chemical Information and Modeling 62.19 (2022): 4810-4819.

---

> > ### Author Response · Authors · 2024-11-19
> >
> > ## Q: Concern about the similarity based score will affect the search of de novo drug molecules.
> >
> > This question is of great value. Firstly for FDA drugs, we believe that the 2582 molecules currently approved by the FDA as small-molecule drugs possess certain favorable drug-like chemical properties, such as stability, activity in the human body, toxicity profile, absorption, and proper metabolism. These properties characterize a distribution within a chemical space, and the distance to that chemical space can characterize the possibility a new molecule to be modifide to a drug. **If a model consistently generates small molecules across various targets that are far from the known drug-like space, the model’s ability to generate drug-usable molecules may be questionable.**
> >
> > Similarity to the known actives is a similar rationale. **Our metrics are designed to evaluate whether a model has the capability to generate actives, rather than to serve as optimization objectives guiding the model to produce only molecules similar to actives.** Our metric assesses the maximum similarity across multiple generated molecules. If a model is unable to generate any molecule similar to actives, then its capability should be questioned.
> >
> > Overall, **we would like to clarify that our proposed metrics are designed to evaluate the effectiveness of current generative models. In other words, our framework is intended for evaluation at the model level, rather than the individual molecule level, and is not meant to serve as optimization objectives during model training or sampling. Optimizing directly for these metrics would not be meaningful. As an evaluation method, similarity-based metrics will not affect the search of de novo drug molecules**
> >
> > In this work, the similarity metric serves as an auxiliary measure, with the primary metric being virtual screening, which directly evaluates the effectiveness of the generated molecules when used for ligand-based virtual screening. Of course, this metric is imperfect. We do not intend for the similarity metric to replace the theoretically estimation-based metrics like Vina docking score or QED, but rather aim to provide an additional perspective for supplementary assessment beyond those scores.
> >
> > ## Q: Docking pose images of molecules generated by MolCRAFT and Pocket2Mol exhibit good similarity to known active.
> >
> > Thanks for your comments. We have supplemented our paper with pictures showing the docking poses of generated high-similarity molecules and the known actives. Please refer to **Appendix C.2** of the updated version of the paper, it can be observed that similar fragments generate identical interactions with the target. If you have any further questions, feel free to let us know.
> >
> > Those images can also be found in the anonymous link: https://anonymous.4open.science/r/tmp-23B1/

---

> > > ### Author Response · Authors · 2024-11-26
> > >
> > > Dear Reviewer,
> > >
> > > Thank you very much for your time and effort in reviewing our paper and rebuttals. We are encouraged to see that our clarifications, along with the additional experimental results and figures, have addressed your concerns and questions, as reflected in your revised score. We are more than willing to provide further clarifications if anything remains unclear.
> > >
> > > Thank you once again for your constructive and insightful feedback, which has been invaluable in helping us refine our paper.
> > >
> > > Best regards,
> > >
> > > The Authors

---

> > > > ### Comment · Reviewer_WYJC · 2024-11-26
> > > >
> > > > Dear Authors,
> > > >
> > > > Thanks for addressing my concerns. My rating has raised to 6

---

### Official Review · Reviewer_pELf · 2024-10-29

**Soundness:** 3
**Presentation:** 2
**Contribution:** 2
**Rating:** 6
**Confidence:** 4

**Summary:**

This paper draws attention to pitfalls in evaluation metrics commonly used to assess generative models for structure-based drug discovery. While the authors focus their discussion on 3D methods, the pitfalls discussed extend to any generative model for drug discovery. The authors first highlight the problem with the commonly used docking score metric and how it can be exploited. Next, three sets of evaluation tasks are proposed: delta docking score (assess specific binding), similarity metrics (to known actives and FDA-approved drugs), and virtual screening (can a similarity search from generated molecules identify known actives?). The authors provide insights on the current performance of 3D SBDD generative models and highlight shortcomings of current model development to guide future research.

**Strengths:**

* The authors explicitly highlight pitfalls of current docking score evaluations of generated molecules
* Performance across several 3D models is compared and contrasted
* The rationale is clear when proposing new metrics

**Weaknesses:**

Overall, I agree there should be better metrics for SBDD assessment. However, many of the metrics and challenges proposed by the authors has been extensively reported in existing literature. There are also many existing works that are not 3D-based diffusion/flow models for SBDD that account for the limitations expressed by the authors. As a result, some of the limitations are limitations of the problem formulation and not intrinsically to SBDD metrics. I will organize the comments under each proposed metric.

### **Docking and Synthesizability**
* The authors focus on SBDD generative models but there is a lot of work in this field that does not involve geometric models that can perform SBDD and recent benchmarks also show that these 1D/2D methods outperform 3D methods [1]. The reason this is brought up is because in these works, the limitation of AutoDock Vina to exploit low QED molecules and large molecules is well documented. As a result of this, there are many papers that explicitly optimise docking jointly with QED (and often also with SA score). Here are a few works that do this, many of which have been published at conferences [2, 3, 4, 5]. Therefore, while I do think that many papers are overfitting AutoDock Vina by designing large molecules with low QED, many recent works have acknowledged this and are proposing objective functions that explicitly account for QED.
* A large reason the Delta score is poor for AutoDock Vina is that it is too crude. This is likely to diminish if one uses Glide docking, as the authors implicitly show with the enrichment. The goal could also be explicitly to dock well to 1 protein and not to another protein as was done here [6].
* It is true that metrics like SA score are imperfect to predict synthesizability but it can be well correlated [7, 8].
* Taking Tanimoto similarity to known molecules does not necessarily mean it is easier to synthesize. Synthesizability is complex and single atom changes can turn something synthesizable to something that is not.

### **Similarity Metrics (including Virtual Screening)**
* If a metric is introduced that rewards similarity to known molecules, then the optimisation objective could simply be similarity to these molecules. This has also been shown explicitly in [9] where the model generates molecules similar to a known active while getting good docking scores. In my opinion, we do not necessarily want to re-discover known molecules in the sense of generating similar molecules to a set of reference ligands. This is already done in distribution learning/transfer learning approaches that were detailed in this review [10]. The alternative and more general goal is to design molecules with arbitrary property profiles. In this case, the degree to which optimisation is successful is the most relevant metric.
* Similarity to FDA-approved drugs may not be completely justified. Drugs are specific and drugs targeting 2 different proteins can be vastly different in structure. In Table 3, the authors also show that even the reference ligand have low similarity to the FDA-approved drugs.
* The problem with Vina scores that the authors elude to is that it is too crude. A very well known problem of docking algorithms is that they cannot reproduce experimental binding poses. Specifically, re-docking is a common task used to validate docking. In cheminformatics literature, this is known as sampling power [11]. Similarly, the ability to rank affinity with docking scores is known as scoring power.
* In ML literature, an equivalent is the assessment of how many binding poses are within some RMSD threshold, for example in the DiffDock paper [12]. If the docking cannot reproduce known experimental data, it is unclear the level of confidence to trust the output. The reason for bringing this up is because if there is a docking algorithm that can capture the binding interactions properly, designing molecules optimised for docking score can implicitly capture similarity. This was explicitly shown in cheminformatics literature [13] (although I still believe similarity to known molecules is not necessarily the best metric and can be hacked). The idea of having a better docking algorithm equating to better results is also shown by the authors in Table 1: comparing Glide with AutoDock Vina and Glide gets better enrichment. Then if Glide were optimised instead, the generated molecules would be better. I understand that this is not commonly done because Glide is proprietary software.

While I believe the sentiment of the authors is that better metrics can better reflect real impact, metrics do not replace wet-lab experiments. This line should not be bolded to give the false impression that predictive methods are close to accurately predicting wet-lab experiment. It would be better to re-phrase this line to convey that better metrics can better inform future model development that may improve wet-lab outcomes.

### **References**
[1] 3D SBDD Benchmark: https://chemrxiv.org/engage/chemrxiv/article-details/66bb0911a4e53c48763ac057

[2] Reinforced GA: https://arxiv.org/abs/2211.16508

[3] TacoGFN: https://arxiv.org/abs/2310.03223

[4] GEAM: https://arxiv.org/abs/2310.00841

[5] Saturn: https://arxiv.org/abs/2405.17066

[6] Negative Design: https://iclr.cc/virtual/2023/12911

[7] SA Correlation 1: https://jcheminf.biomedcentral.com/articles/10.1186/s13321-023-00678-z

[8] SA Correlation 2: https://arxiv.org/abs/2407.12186v1

[9] Similarity reward: https://www.nature.com/articles/s42256-022-00494-4

[10] Generative Design Review: https://www.nature.com/articles/s42256-024-00843-5

[11] Sampling and Scoring Power: https://pubs.rsc.org/en/content/articlelanding/2016/cp/c6cp01555g

[12] DiffDock: https://arxiv.org/abs/2210.01776

[13] DockStream: https://jcheminf.biomedcentral.com/articles/10.1186/s13321-021-00563-7

**Questions:**

* What are the authors' thoughts on evaluating the 3D geometry of the generated molecules? PoseCheck [1] and PoseBusters [2] are now common in SBDD papers. The results show that unphysical poses is an ongoing challenge for 3D generative models. Generated molecules with poses that clash with the protein would not typically be considered further.

* The authors state that the metrics do not require the need to consider synthetic ability but synthesizability is a problem highlighted in the Methods section. I agree that wet-lab testing should never be a requirement for computational research but do the authors think there are metrics that can better assess synthesizability beyond SA score and beyond being implicitly considered via similarity to known actives?

* Minor comment: there is a typo in "actives" in figure 2.

[1] https://arxiv.org/abs/2308.07413

[2] https://pubs.rsc.org/en/content/articlehtml/2024/sc/d3sc04185a

---

> ### Author Response · Authors · 2024-11-19
>
> Thank you for taking the time to read our manuscript and for providing detailed review comments. Thank you for acknowledging our perspective on the need for improved metrics in SBDD. We have carefully considered your perspectives on the SBDD metrics. To provide you with a more comprehensive understanding of our work and minimize potential misunderstandings, we will first **introduce the motivation** behind our proposed set of metrics. Following that, we will respond in detail to your **discussion and viewpoints on each type of metric**. Finally, we will offer some **reflections on the questions** you raised. We hope this information helps to further clarify our work, we look forward to engaging in further discussions with you.
>
> # 1.Further explanation of our motivation:
>
> **First and foremost, we do not intend for the proposed metrics to entirely replace existing ones, nor do we expect that introducing and optimizing a single perfect metric would solve the challenges in the SBDD field.** Instead, our goal is to provide an additional evaluation perspective, which stems from our consideration of how SBDD models can be practically applied. We want to
>
> Currently, the primary method for deploying SBDD models is synthesizing the generated molecules for wet-lab experiments. However, the synthesis of these molecules and the wet-lab experiments themselves are very costly. And those generated molecules that cannot be synthesized are treated as useless. We believe that beyond using SBDD-generated molecules directly as drug candidates, **they can be used in more practical ways**. One such application is using the generated small molecules as templates for ligand-based virtual screening, where similar compounds from chemical libraries are identified and tested as drug candidates in wet-lab experiments. Additionally, human experts can leverage their domain knowledge to modify and improve upon the generated molecules. **Although some generated molecules cannot be synthesized, they can still be effective in drug discovery pipeline**. We believe these deployment strategies for SBDD models have its value in pharmaceutical contexts, providing information on potentially useful compounds or functional groups. Furthermore, given the economic costs and challenges associated with synthesis, these approaches may be more practical and feasible for real-world implementation.
>
> **We hope to provide a more comprehensive evaluation of whether current SBDD models are useful and effective in the whole drug discovery pipeline.** However, current evaluation metrics primarily focus on assessing the quality of generated molecules as a drug candidate directly, without considering other deployment strategies for SBDD models. We aim to fill this blank by proposing metrics that can evaluate model performance in these alternative, yet impactful, applications.
>
> There always exists generated molecules that cannot be synthesized for experimentation, making them useless in that context. However, these molecules can still be useful as they may **contain good fragments or substructures**, and our metrics can assess this usefulness. Therefore, our evaluation provides **a broader perspective for assessment**.
>
> **Fortunately, we have observed that some recent wet-lab studies have already begun deploying SBDD models in these ways**, reinforcing the importance of developing evaluation metrics for these applications. For example, the TamGen team [1] sought commercially available compounds similar to those generated by their model from a 446k compound library. They successfully identified 159 analogs, and five of these analogs demonstrated significant inhibitory effects in a wet-lab ClpP1P2 peptidase activity assay. Similarly, the Pocket Crafter project [2] employed SAR enrichment analysis to extract valuable scaffolds from the generated molecules, using these scaffolds to conduct ligand-similarity-based virtual screening. They done wet-lab experiment on the 2029 compounds searched from  Novartis internal archived diverse library and led to the finding of WM-662 and Compound 1 [3], the known WBM pocket binding molecules.

---

> > ### Author Response · Authors · 2024-11-19
> >
> > The **core metric we propose is the virtual screening metric**, which directly evaluates the effectiveness of model-generated molecules when used for ligand-based virtual screening. Supporting this is the similarity metric to known actives, which assesses the ease with which human experts can modify the generated molecules into practical and usable compounds.
> >
> > **We want to emphasize that these metrics are designed to assess whether a model is effective in a more direct way**; however, we do not intend for them to be the optimization objectives for reinforcement learning or other optimization algorithms,  Pharmaceutical challenges in real-world scenarios are highly complex and require the satisfaction of multiple objectives, making them quintessential multi-objective optimization problems. Just as optimizing solely for Vina scores can lead to models being exploited by generating excessively large molecules, or solely optimizing for QED can result in the generation of overly small molecules, our metrics are also not meant to be optimized in isolation. Instead, **we hope that our metrics can provide a fair assessment of whether a model can be useful in the drug discovery pipeline**
> >
> > We fully agree with your statement that "metrics cannot replace wet-lab experiments". When generated molecules are used directly as drug candidates, existing metrics, including various docking scores, although have some kind of correlation, do not directly reflect the outcomes of wet-lab experiments. This underscores the need for designing better metrics, such as improved docking scores. **However, when we shift our focus to a different point of view that does not assume that the generated molecules must be used directly, we can identify metrics that better align with experimental results.** DUD-E, for instance, is a well-validated benchmark for wet-lab virtual screening, and the difference between generated molecules and real actives indeed reflects the difficulty experts face in modifying generated molecules into real actives. We believe that directly using generated molecules as drugs remains a crucial and efficient goal and represents the ultimate aim of SBDD, though it still requires further research to bridge existing gaps. **In the meantime, we believe that proposing a set of evaluation metrics for other, already practiced deployment methods also holds significant value.** We hope this explanation clarifies our motivations.
> >
> > [1] Wu, Kehan, et al. "TamGen: drug design with target-aware molecule generation through a chemical language model." Nature Communications 15.1 (2024): 9360.
> >
> > [2] Shen, Lingling, et al. "Pocket Crafter: a 3D generative modeling based workflow for the rapid generation of hit molecules in drug discovery." Journal of Cheminformatics 16.1 (2024): 33.
> >
> > [3] Ding, Jian, et al. "Discovery and structure-based design of inhibitors of the WD repeat-containing protein 5 (WDR5)–MYC interaction." Journal of Medicinal Chemistry 66.12 (2023): 8310-8323.

---

> > > ### Author Response · Authors · 2024-11-19
> > >
> > > # 2.Responses to your valuable comments and perspectives
> > >
> > > Thank you for sharing your insights with us. We will address each point in detail.
> > >
> > > ##  Docking and Synthesizability
> > >
> > > ### 2.1 Regarding the metrics optimization problem
> > >
> > > Thank you for highlighting that many papers focus on jointly optimizing QED and Vina scores. **While our goal is not to replace current metrics, we still aim to highlight the necessity of exploring metric design from additional perspectives by analyzing the issues associated with existing metrics.**
> > >
> > > We agree that such optimization approaches can generate molecules with high QED, Vina, and even SA scores. However, the issue with these theoretical estimation-based metrics is that, while they correlate to real-world outcomes to some extent, they are still **susceptible to exploitation during optimization**.
> > >
> > > For instance, we tested an optimization-based method asked by another reviewer. After five rounds of optimization, the Vina docking score improved from -9.00 to -9.66. However, when we calculated the Glide scores for the same molecules, the results actually worsened, shifting from -6.20 to -6.02. This suggests that the optimization process may not improve the quality of the generated molecules but instead exploits the scoring function itself.
> > >
> > > While more accurate metrics—such as transitioning from Vina to Glide, MMGBSA, or even FEP—might mitigate this issue, **currently these metrics still cannot guarantee that improved scores reflect genuine enhancements in the model’s ability to generate better molecules**. Instead, the model might simply learn to generate molecules favored by those specific scoring functions. Similarly, multi-objective optimization can indeed produce molecules with good docking and QED scores. However, it remains challenging to determine whether these high scores genuinely indicate high-quality molecules or merely reflect alignment with the biases of the scoring functions. This concern becomes even more evident when a molecule optimized for one scoring function performs poorly under a different scoring function. **Thus, solely relying on these metrics can lead to overly optimistic assessments of current SBDD models. For example, many models claim to generate 50% of molecules that outperform the reference ligand or achieve a 25% success rate in meeting all the multiple objective properties they define. If these claims were truly reflective of reality, drug discovery would already be a solved problem. By now, we should have produced hundreds or even thousands of useful drugs using SBDD models. But is that really the case?**
> > >
> > > Returning to our **benchmark**, our motivation is to evaluate the usefulness of generative models **from a different perspective, rather than solely relying on theoretical estimation-based metrics.** Our intention is **not to replace** original metrics like Vina and QED but to propose a multi-faceted evaluation framework. For binding affinity estimation, we include the Delta Score, which measures whether high docking scores are due to overfitting or genuinely specific to different targets. We also introduce the DrugCLIP score, which demonstrates superior scoring power compared to docking scores, including Glide. Furthermore, we provide a **virtual screening-based metric**, which is the cornerstone of our evaluation framework. The advantage of virtual screening is that all active and inactive molecules used are wet-lab validated, making the entire process a simulation of real-world scenarios and directly correlated with the success rates of wet-lab experiments.
> > >
> > > In summary, while multi-objective optimization methods can produce molecules that perform well across different metrics, we believe it is essential not to rely solely on theoretical estimation-based metrics. **Incorporating evaluations from additional perspectives would be beneficial.**

---

> > > > ### Author Response · Authors · 2024-11-19
> > > >
> > > > ### 2.2 Regarding Delta score
> > > >
> > > > We are uncertain about the intended meaning of “Delta score is poor for AutoDock Vina.” However, we are open to providing further clarification. Our use of the delta score is intended to evaluate the models and not to imply any inaccuracy on the part of Vina. **In fact, the delta score results reported in our paper were calculated using Glide. We will add further annotations in the manuscript to clarify this.** As you mentioned, the calculation involves measuring the average difference in docking scores between one protein and another. We may not fully understand the reviewer’s concerns, but we are willing to provide additional explanations if there is still any unclarity in our paper.
> > > >
> > > > ### 2.3 Regarding the SA score is correlated to synthesizability
> > > >
> > > > We agree that SA scores are correlated with a molecule’s synthesizability. However, the issue is that, although many molecules can achieve a good SA score, people with biochemistry backgrounds in our team often find these molecules difficult or expensive to synthesize. Motivated by this, our approach seeks to move beyond the traditional mindset that the generated molecule must be directly synthesized for wet-lab experiments. Instead, we aim to evaluate the usefulness of generative models without considering the synthesizability of the molecules themselves. This is why we propose metrics based on similarity to known actives and virtual screening-based metrics. **Overall, we hope the reviewer understands that we do not dismiss the value of SA scores. Rather, we present a set of metrics unrelated to the inherent synthesizability of the molecule, focusing instead on evaluating SBDD models through alternative deployment methods such as virtual screening.**
> > > >
> > > > ### 2.4 Regarding Synthesizability is sensitive
> > > >
> > > > We completely agree with your point that similarity to known molecules does not necessarily imply ease of synthesis. We apologize for any confusion caused by our paper, but we want to clarify that **we never made this claim**. Our point is that we do not need to consider whether the generated molecules can be directly synthesized or not.
> > > >
> > > > As you mentioned, even a single atom change can render a molecule synthesizable or not. This means that if a generated molecule is similar to a potential active compound that is in stock or easy to synthesize, we can either modify the generated molecule to the potential active compound or perform virtual screening to identify it. In this way, the generated molecule remains useful without needing to be directly synthesizable itself.

---

> ### Author Response · Authors · 2024-11-19
>
> ## Similarity Metrics (including Virtual Screening)
>
> ### 2.5 Regarding the similarity metrics
>
> **Firstly, we would like to clarify that our proposed metrics are designed to evaluate the effectiveness of current generative models. In other words, our framework is intended for evaluation at the model level, rather than the individual molecule level, and is not meant to serve as optimization objectives during model training or sampling. Optimizing directly for these metrics would not be meaningful.**
>
> In machine learning, if there is an explicitly defined objective or reward function, it is almost certain that models, with proper optimization methods, can generate molecules that satisfy those specific properties or objectives. However, our focus is on evaluating models, not directly optimizing for these metrics. Indeed, not all evaluation metrics in machine learning are suitable to be used as optimization targets.
>
> Additionally, our goal is not to **re-discover known molecules**. As a **benchmark**, our goal is to **test** whether the models **have the ability** to generate active molecules. The ability to generate molecules similar to known actives is intended as a **simulated test** for the generative models. **During sampling, the model does not have access to the information of known actives.** If, under these conditions, the model can generate molecules similar to active compounds or produce references that can be used for virtual screening to identify actives, it demonstrates the model’s potential to generalize to **new targets that have not been thoroughly studied or have no known actives. For new targets without any known actives, such a model could still generate molecules that similary to their potential actives and help identify potentially active compounds.**
>
> For the dataset we used for testing, we are not limited to using the reference ligand as the active molecule. Instead, for each target, there is an **average of 224 active molecules, with an average pairwise similarity of approximately 0.2.** We believe this constitutes a **diverse and abundant enough library** to encompass different binding molecules. **On the other hand, if a model lacks the ability to generate molecules similar to or capable of retrieving at least one of these actives, then its generative power should be called into question.**
>
> In conclusion, our framework is not about forcing the model to re-discover known molecules. Instead, it is about evaluating whether the model has the capability to identify active molecules and provide valuable starting points for drug discovery.
>
>
> ### 2.6 Regarding similarity to FDA approved ligand
>
> We believe that the 2582 molecules currently approved by the FDA as small-molecule drugs possess certain favorable drug-like chemical properties, such as stability, activity in the human body, toxicity profile, absorption, and proper metabolism. **These properties characterize a distribution within a chemical space**, and whether the molecules fall into that chemical space can characterize the possibility a new molecule can also be a drug. We agree that drugs are specific and drugs targeting 2 different proteins can be vastly different in structure, but here the similarity to FDA drugs is not a protein-specific metric.
>
> However, we also acknowledge that similarity to known FDA-approved drugs can be a too strict requirement. It is more from a point that to predict whether the generated molecule can ultimately be a real drug, which may not be the scope of SBDD models. **This is not the core of our evaluation metrics, and we are also open to moving it to the appendix section if needed.**
>
> ### 2.7 Regarding reviewer's "similarity to known molecules is not necessarily the best metric and can be hacked"
>
> We agree that similarity to known molecules is not necessarily the best metric. However, we are uncertain why it would be considered easy to hack. It is important to emphasize that we are not solely relying on similarity to known actives. We also provide virtual screening metrics, which evaluate whether the generated molecules are more similar to active compounds than inactive ones.
>
> Regarding concerns about hacking these evaluation metrics during training or sampling, we believe this is unlikely. **Following the evaluation process, there should be no information leakage about the known actives, meaning the similarities or virtual screening abilities cannot be precomputed or exploited.**

---

> > ### Author Response · Authors · 2024-11-19
> >
> > # 3.Reflections and responses to your questions
> >
> > ## 3.1 Regarding the question on evaluating 3D geometry of generate molecules.
> >
> > This is an excellent question. We have been aware of the PoseCheck benchmark for some time and have previously considered and discussed its implications. The answer to this question ties directly into the purpose of using 3D generative models for SBDD.
> >
> > Regarding the question, benchmarks like PoseCheck can serve as valuable references to test whether a 3D SBDD model learns an accurate 3D representation and can generate realistic conformations. If a model effectively captures a 3D representation of protein-ligand complexes, it should avoid producing unphysical poses or molecules that clash with protein pockets in 3D space. However, this does not necessarily indicate that the generated molecule is useful or not. A molecule may still contain valuable fragments or functional groups, even if its initial 3D conformation is invalid or contains clashes. We view molecule generation and docking (binding pose prediction) as separate tasks.
> >
> > In summary, geometry-based evaluation is a useful tool for assessing a model’s ability to learn 3D representations and generate valid 3D conformations, which is indeed very important and useful for docking task. However, in terms of generation tasks, it is not sufficient for evaluating the model’s capacity to generate functionally useful molecules, which is the ultimate goal for SBDD. Also, such evaluation cannot be applied to 1D/2D generative models. Filtering out molecules that clash with the protein may not be ideal either, as these molecules could still hold potential and may yield valid conformations after re-docking or in real-world testing. Our proposed metrics adopt a more practical perspective, focusing on whether generated molecules contain critical structural elements or fragments that may lead to success in wet-lab experiments. Thus, we believe our metrics complement the 3D geometry-based metrics in PoseCheck: while PoseCheck focuses on a model’s understanding of 3D geometry, our metrics focus on the SBDD model’s ability to generate useful molecules.
> >
> >
> > ## 3.2 Regarding the question on evaluating the synthesizability of generated molecules.
> >
> >
> > Thank you for raising this insightful question about synthesizability. **We would like to clarify again that we are not using similarity to known active compounds as a metric for synthesizability.** We recognize that even similar molecules can vary significantly in terms of their ease of synthesis. Our point is that if a molecule is structurally similar to a known active compound, it suggests that the generated molecule may have potential for modification by experts to become an active molecule, or that it contains fragments useful for binding to the target.
> >
> > To address the question directly, we believe that purely theoretical metrics are insufficient for evaluating synthesizability. Synthesizability is inherently complex and cannot be adequately captured by a single scalar metric like SA score. A more practical approach is to conduct a detailed retrosynthetic analysis to determine whether a reasonable synthesis pathway can be proposed. Such analysis can be performed using software tools or machine learning models, and we view this as another essential task for SBDD. Another reliable method should be using human chemistry experts for evaluation.
> >
> > Lastly, we want to emphasize again that our evaluation framework actually provide a systematic solution to bypass the problem in the synthesizability of molecules generated by SBDD models, since we do not assume them to be synthesizable to be useful in the drug discovery pipeline.
> >
> > # 4
> >
> > We would like to thank the reviewer again for reading our paper and providing valuable feedback. **We hope that our explanations above can address some of the reviewer’s concerns. We warmly welcome continued discussions with the reviewer.**

---

> ### Comment · Reviewer_pELf · 2024-11-20
> **Response to authors**
>
> First of all, thank you for the detailed replies and the effort put into the rebuttal. Some points have been clarified. However, I have follow-up questions and concerns that ultimately are centred around the novelty of the proposed metrics and use cases. There have been many papers in cheminformatics journals that have addressed and commented on certain points raised by the authors. As a result, I believe many limitations are a result of the optimization objectives used at machine learning conferences. I will follow the same points to reply.
>
> ### 1. Further explanation of our motivation
>
> Thank you for clarifying the intended use case of the proposed metrics and I apologize if I misinterpreted. The response from the authors in this section is centred on the proposition that generated molecules can be useful even if not directly used. The authors cite TamGen and Pocket Crafter which either use information from the generated molecules or look for similar analogues in commercial libraries for experimental validation. I agree that this is a valid use case but this has been done for decades. Before TamGen and Pocket Crafter and before machine learning was popular in drug discovery, cheminformatics workflows already use similarity based on reference molecules to find other hits. In the context of generative models, this has been done in these papers [1, 2] for example. In the CACHE challenge, this is another example of using a generative model's output to find analogues [3]. This is also very similar to transfer learning to use known actives to generate similar analogues (in some sense similar to analogue searching). This review paper detailed examples of such papers that have showed experimental validation [4]. I appreciate the authors' motivation in proposing useful metrics to assess generative models but the idea of analogue searching is something that has existed for a long time.
>
> ### 2. Docking/synthesizability: Regarding the metrics optimization problem
>
> In the new optimization-based method the authors performed, they state that the Vina docking scores improved but Glide scores worsened. This is not entirely surprising if Vina and Glide model different things and Vina is often much more crude of an affinity estimator. If the optimization were performed with Glide instead, then the Glide scores would not worsen. This is a limitation of the docking algorithm.
>
> The authors next state that going from Vina to Glide, MMGBSA, FEP still does not guarantee better molecules. Models that get better scores than the reference ligand are not necessarily successful molecules. I agree with these statements but this is exactly a limitation of the predictive ability. I appreciate this is where the authors are trying to propose extra metrics for, but in practice, these binding affinity estimates are rarely used as is, as is done in most machine learning papers. Here is an example of a paper showing that many docking algorithms are not even correlated with binding affinity [5]. Here is an example of a paper where a better docking algorithm can give better FEP estimations [6]. My intended message here is that the problem with the metrics used  is that they are not validated beforehand. The virtual screening metric proposed by the authors can be used to retrospectively validate models but what about for any arbitrary protein? In the end, I believe the molecules generated from SBDD models will need to be tested with FEP or something similar. Regarding models not generating molecules similar to known actives, this is handled by validating that the optimisation objective is modelling the correct dynamics. I refer again to the review paper for all the experimentally validated SBDD methods [4]. Here is also an example where this is explicitly showed [5]. Therefore, my concern is that the proposed metrics do not remove the need for validating optimisation objectives for prospective discovery. As a result, I think there should be more encouragement on validating the optimisation objectives instead. **Can the authors share their thoughts about this? I am happy to continue the discussion.**
>
>
> [1] https://pubs.acs.org/doi/10.1021/acs.molpharmaceut.7b01137
>
> [2] https://pubs.acs.org/doi/full/10.1021/acs.jcim.3c01818
>
> [3] https://cache-challenge.org/challenges/app/61f7e09526539
>
> [4] https://www.nature.com/articles/s42256-024-00843-5
>
> [5] https://jcheminf.biomedcentral.com/articles/10.1186/s13321-021-00563-7
>
> [6] https://www.nature.com/articles/s42004-023-00859-9

---

> > ### Comment · Reviewer_pELf · 2024-11-20
> > **Response to authors**
> >
> > ### 2.2 Regarding Delta score
> >
> > I apologize for the vague comment. I was wondering what the delta score would be if Glide was used for constrained docking. A big advantage of Glide is its ability to enforce specific interactions in the output poses, which affect the score. Most binders form some specific interactions with proteins. I was wondering if this was done, that the docking scores would be much better in distinguishing between different targets.
> >
> > ### 2.3/2.4/3.2 Regarding the SA score/synthesizability
> >
> > The authors have clarified my question on SA score.
> >
> > ### 2.5 Regarding the similarity metrics
> >
> > Thank you for the clarifications. I do see value in the diverse actives set for SBDD model assessment. I am currently thinking more about the authors' response. One thing that would be useful is to provide statistics on the "actives". What are their measured affinities? It would also be interesting to partition the actives into different potency ranges.
> >
> > ### 2.6 Regarding similarity to FDA approved ligand
> >
> > I do still believe the similarity to FDA molecules is too strict. I appreciate this is not a main metric proposed by the authors.
> >
> > ### 2.7 Regarding reviewer's "similarity to known molecules is not necessarily the best metric and can be hacked"
> >
> > Thank you for the clarification. I agree there should be no information leakage. The only comment I want to leave is to caution that many SBDD papers I have seen make comparisons when training models on different subsets of data. If certain subsets contain the protein and/or homologs, it would not be a fair comparison against different SBDD models. Since the authors are proposing metrics, it would be useful to highlight this issue of training on different subsets of data.
> >
> > ## 3.1 Regarding the question on evaluating 3D geometry of generate molecules
> >
> > Thank you for the answer. I agree with the authors' response.
> >
> > Thank you for the detailed responses from the authors. Overall, I am open to raising the score following further clarifications.

---

> > > ### Author Response · Authors · 2024-11-22
> > >
> > > # Official Comment
> > >
> > > Thank you for taking the time to read our response. We are glad that our previous reply was able to address some of your concerns. We also appreciate you sharing your further thoughts with us. Below, we provide our responses to your other concerns.
> > >
> > > # 1. Further explanation of our motivation
> > >
> > > Thank the reviewer for providing additional examples of deploying SBDD models through ligand-based virtual screening or analogue searching. Here we aim to minimize potential misunderstandings. We agree that these deployment strategies have been proposed long ago. However, we did not claim in our paper that these deployment methods were introduced by us. Instead, we introduced **a new benchmark** that evaluates the performance of models under these deployment strategies as computational evaluation metrics.
> > >
> > > We want to emphasize that, **as a benchmark, our core lies in proposing a set of metrics to assess SBDD models**, rather than exploring new deployment methods for these models. To best of our knowledge, **we are the first to systematically evaluate the performance of current mainstream 3D-SBDD models under these deployment strategies and provide standardized settings including training and testing datasets.** These aspects constitute the novelty of our work as a benchmark.
> > >
> > > Precisely because many researchers deploy SBDD models using these strategies, we are even more convinced that these deployment methods are gaining attention. **This further supports the idea that evaluating a model’s performance under these deployment strategies should be considered an integral aspect of a comprehensive evaluation of SBDD models.** We sincerely thank the reviewer for introducing additional works utilizing virtual screening and analogue searching in wet-lab pipelines. We will incorporate these examples into our paper to further strengthen the rationale for the metrics we propose.
> > >
> > > # 2.Docking/synthesizability
> > >
> > > ## 2.1 Regarding the metrics optimization problem
> > >
> > > We thank the reviewer for sharing some previous works with us, and we are more than happy to share our thoughts on these issues. From our understanding, the reviewer’s concerns primarily focus on two aspects:
> > >
> > > - The generalizability and reliability of the virtual screening metric.
> > >
> > > - Proposing new metrics does not eliminate the need to validate existing optimization objectives.
> > >
> > > We will address these two points in detail below.
> > >
> > > ### The generalizability and reliability of the virtual screening metric
> > >
> > > This aims to address the reviewer’s concern regarding whether the model can generalize correctly to unseen data. In fact, the generalizability of virtual screening has already been **validated both computationally and through wet-lab experiments.**
> > >
> > > Computationally, the training set **does not include any** proteins that are structurally similar to those in the virtual screening test set, as will be further detailed in the subsequent **response to 2.7**. Thus, the targets in the test set are **unseen new proteins for the model**. Our metric is to measure the model’s performance in virtual screening for these new targets. Therefore, if a model performs well on our virtual screening metric, it already demonstrates the model good at conduct virtual screening on previously unseen target. The DUD-E test set we used includes **over 100 targets distributed across 8 categories**, including Kinase, Protease, GPCR, Ion Channel, Nuclear Receptor, Cytochrome P450, Other Enzymes, and Miscellaneous. The test results on these over 100 unseen targets provide a reasonable level of confidence in the generalizability of our approach. We think it is a **standard setting** to test the generalization ability of models in machine learning domain.
> > >
> > > In terms of wet-lab validation, **several studies have demonstrated an alignment between virtual screening metrics and wet-lab performance.** For instance, [1] and [2] proposed virtual screening methods that achieved high scores on DUD-E and LIT-PCBA benchmarks while in wet-lab experiments, [1] successfully identified non-covalent inhibitors for the challenging protein target GPX4, which has a flat surface, with an IC50 of 4.17 μM. Meanwhile, [2] identified 12 diverse molecules with significant affinities for NET, a newly solved protein structure. It achieved a hit rate of 15% in the wet-lab experiments which matched the computational metric estimation.
> > >
> > > However, the issue of generalizability can only be mitigated, never fully resolved. No metric can ensure that a model performing well on a fixed test set will also perform consistently across all new data, regardless of the machine learning task.
> > >
> > > [1] Wang, Zhen, et al. "Enhancing Challenging Target Screening via Multimodal Protein-Ligand Contrastive Learning." bioRxiv (2024): 2024-08.
> > >
> > > [2] Jia, Yinjun, et al. "Deep contrastive learning enables genome-wide virtual screening." bioRxiv (2024): 2024-09.

---

> > > > ### Author Response · Authors · 2024-11-22
> > > >
> > > > ### Proposing new metrics does not eliminate the need to validate existing optimization objectives.
> > > >
> > > > We fully agree with the reviewer’s point that better-validated optimization objectives are essential and should be further researched. However, we believe that as an evaluation benchmark, rather than an optimization objective, our work still holds value. Evaluation metrics and optimization objectives are two distinct concepts. Ideally, they could align, but this is not always the case. **Our goal in proposing these evaluation metrics is to systematically evaluate and compare current models from new perspectives and to gain insights from these comparisons to guide the development of future models.** This guidance does not necessarily mean introducing a metric that new models must improve upon. Instead, analyzing how existing methods perform on a given metric can also provide valuable insights to the community. The rapid advancements in SBDD underscore the critical need for rethinking and evaluating the actual effectiveness of generative models. A diverse array of models has emerged, ranging from VAEs with voxel grids, to GNNs operating in an autoregressive manner, to diffusion-based approaches, flow matching models, and Bayesian Flow Network-based models. It’s crucial to identify which foundational generative models that can accurately capture the distribution of binding protein-ligand pairs. Without this critical analysis, the field risks advancing models that may appear effective based on misleading or insufficient metrics but lack true reliability. This could ultimately impede the progress of SBDD research and development. That is why we think our benchmark is essential to address this challenge. Its value lies in a multifaceted evaluation framework that identifies models with real effectiveness, enabling the community to optimize and build upon these reliable foundations. Such benchmarks play a crucial role in ensuring that advancements in SBDD are grounded in robust, validated models, thereby driving meaningful and sustainable progress in the field.
> > > >
> > > > We strongly agree with the reviewer that current metrics, such as MMGBSA and FEP, which provide better approximations of binding affinity, are not widely utilized in machine learning papers. A key reason for this is the substantial computational cost associated with these metrics. In machine learning model evaluations, a large number of molecules must be generated and evaluated across many targets to ensure statistically robust results and conclusions, which leads to significant computational demands. As far as we know, obtaining an FEP score for a single protein-molecule pair requires approximately 7 A100 GPU hours. Evaluating an entire test set could require more than a year of A100 GPU hours per model, which is impractical.
> > > >
> > > > Coming back to your concern, **we firmly believe that our proposed benchmark does not conflict with the pursuit of better optimization objectives**. On the contrary, these two aspects are complementary and mutually reinforce each other. **While our benchmark focuses on evaluating models at a foundational level, better optimization objectives are crucial for guiding molecule generation toward improved properties. Together, they form essential components of SBDD.**
> > > >
> > > > The baseline models considered in our benchmark are purely generative models, trained to learn the distribution of molecules that bind to specific targets. Ideally, evaluation metrics should quantify the difference between the entire space of generated molecules and that of all active molecules for the target. Since the complete space of active molecules is unknown, the use of known active molecules is the most practical approximation. This ensures that our evaluation framework aligns closely with the learning objectives of machine learning-based generative models. Our evaluation logic aligns with commonly used methods in other domains. For instance, language models are often evaluated using metrics like BLEU and ROUGE, which measure the degree of overlap between the generated text and reference texts.
> > > >
> > > > We acknowledge the significance of developing better optimization objectives to bridge the gap between computational predictions and experimental outcomes. **In our perspective, these efforts are complementary: our evaluation framework offers a reliable foundation for assessing how effectively generative models learn the desired molecule space, while improved optimization objectives can build upon this foundation to refine the generation process.** By doing so, optimization objectives can guide the generation of molecules with enhanced properties like binding affinity with the base generative models selected with our evaluation framework.

---

> > > > > ### Author Response · Authors · 2024-11-22
> > > > >
> > > > > We hold deep respect for the reviewer's expertise in this field and sincerely value the insightful questions and perspectives they have shared. This represents a truly valuable and unique opportunity for us to learn and engage in meaningful dialogue about the advancement of SBDD. In this spirit, we would like to also share some thoughts on what constitutes an effective optimization objective. Ideally, the optimization objective would rely on wet-lab experiments, as a lab-in-the-loop pipeline represents the ideal way for drug discovery, as successfully done by Prescient Design. However, given the challenges of conducting wet-lab experiments—particularly for small molecules compared to antibodies—it is worth exploring the integration of human experts in a human-in-the-loop design pipeline. Additionally, more accurate but computationally expensive scoring functions, such as FEP, can be used to simulate wet-lab experiments. However, such methods are limited as they primarily estimate binding affinity, whereas human experts can provide a broader evaluation of the molecules, considering various critical factors.
> > > > >
> > > > > We sincerely thank the reviewer for providing valuable references and examples. We will incorporate some of points from these papers into our manuscript.
> > > > >
> > > > >
> > > > > ## 2.2 Regarding Delta score
> > > > >
> > > > > We thank the reviewer for further elaborating on their question to facilitate deeper discussion. We understand the reviewer’s point is that delta score becomes more meaningful with more accurate docking score methods. We fully agree with that.
> > > > >
> > > > > We used the delta score to quantify **the difference in docking scores for generated molecules when binding to their intended target versus other targets**. A larger delta score indicates that the generated molecule demonstrates a strong docking score specifically for its intended target, rather than uniformly high scores across multiple targets. In cases where the delta score is low, it suggests either that the molecule lacks specificity, making it unsuitable for the intended target, or that the **favorable docking score arises from overfitting to the docking software rather than genuine binding interactions with the target.**
> > > > >
> > > > > Actually, We believe this updated evaluation method enables a more accurate and fair comparison between models by addressing the tendency of docking software to be easily overfitted, thereby reducing the risk of misleading conclusions.
> > > > >
> > > > > In our experiments, we believe that the delta scores calculated using Glide are already able to reflect, to some extent, the difference of the models. As shown in Table 2 of the paper, we measured the delta score for the reference ligand, which significantly exceeded the delta scores of the SBDD models. Furthermore, while the delta scores for the models are relatively low, there are clear differences among the models, and these differences remain consistent across repeated tests. This demonstrates that the models vary in their ability to generate molecules that have specific high docking scores, which gives much more insights than relying on the conventional docking score comparison approach.
> > > > >
> > > > > The constrained docking method mentioned by the reviewer is indeed a more accurate docking approach. However, as far as we know, adding constraints to Glide requires providing human prior knowledge for each target, essentially making it a human-involved evaluation. This limitation makes it infeasible for large-scale evaluations. That said, we are still eager to address the reviewer’s concerns. Therefore, we are trying to re-evaluate the delta scores using Glide XP, the most precise docking software currently accessible to us. Due to the large number of docking tasks and the inherent slowness of Glide XP, the calculations are still in progress. Once completed, we will provide the results immediately. We appreciate the reviewer’s patience and understanding.

---

> > > > > > ### Author Response · Authors · 2024-11-22
> > > > > >
> > > > > > ## 2.5 Regarding the similarity metrics
> > > > > >
> > > > > > Thank you for highlighting the idea of partitioning active molecules into different ranges. This is an excellent suggestion that can further enhance our evaluation process. For the binding affinities of the active molecules, the active molecules are defined as bioactivity values < 1000 nM no matter which assay type used (Ki, Kd, IC50). We can also further split them into more groups based on the bioactivity ranges.
> > > > > >
> > > > > > We believe this idea on partitioning provides an excellent opportunity to benchmark models with similarity-based metrics at varying levels of molecule activity. Thank you again for pointing this out.
> > > > > >
> > > > > > To better present and elaborate on the dataset, We also want to further provide the information on the diverse nature of the active sets for each target.
> > > > > >
> > > > > > The diversity calculate by (1- RDK fingerprint similarity) is shown below. We compare the active sets for targets, and the molecules generated by MolCRAFT and TargetDiff for targets. The mean values are shown here:
> > > > > >
> > > > > > | Method       | Diversity |
> > > > > > |--------------|-----------|
> > > > > > | Active sets  | 0.633     |
> > > > > > | MolCRAFT     | 0.618     |
> > > > > > | TargetDiff   | 0.581     |
> > > > > >
> > > > > >
> > > > > > It is interesting to see that the diversity between the active molecules actually have a better diversity than the molecules generated by models.
> > > > > >
> > > > > > We also provide several distribution plots to demonstrate such diversity:
> > > > > >
> > > > > > logp: https://anonymous.4open.science/r/tmp-23B1/logp.jpeg
> > > > > >
> > > > > > molecule weight: https://anonymous.4open.science/r/tmp-23B1/mw.jpeg
> > > > > >
> > > > > > hydrogen bond acceptors: https://anonymous.4open.science/r/tmp-23B1/hba.jpeg
> > > > > >
> > > > > > hydrogen bond donors: https://anonymous.4open.science/r/tmp-23B1/hbd.jpeg
> > > > > >
> > > > > > ## 2.6 Regarding similarity to FDA approved ligand
> > > > > >
> > > > > > Yes, at this point we also agree that similarity to FDA is too strict evaluation and is not beyond the ability of SBDD models. We will move them to appendix sections.
> > > > > >
> > > > > > ## 2.7 Regarding reviewer's "similarity to known molecules is not necessarily the best metric and can be hacked"
> > > > > >
> > > > > > We completely agree with the reviewer’s statement that the training data must not include proteins from the test set or their homologs to ensure fair comparisons. Due to the importance of this principle, **we constructed our dataset starting from PDBbind and performed a strict split**, instead of use CrossDocked for training. **We remove pockets in the training set that are structurally similar to those in the test set.** Structural similarity was measured using the pocket alignment method FLAPP [1].  We adopted two different FLAPP score thresholds, 0.6 and 0.9, for the splitting process to create two datasets for comparison, aiming to explore the impact of homology on testing performance. **All models evaluated in the paper were retrained on this dataset to ensure fair comparisons.** Details about this process are provided in Section 3.3 of the paper.
> > > > > >
> > > > > > In the future, **we will release the PDBbind FLAPP 60 training set and the PDBbind FLAPP 90 training set to enable other models to train on the same datasets and allow fair comparisons with other methods.** We sincerely appreciate your suggestion and will emphasize in the paper and the public dataset documentation that models must be tested on strictly partitioned datasets to ensure the comparability of the results.
> > > > > >
> > > > > > We also want to **emphasize that this is also one of our contribution to create a fair dataset setting for evaluating different models**.
> > > > > >
> > > > > > [1] Sankar, Santhosh, Naren Chandran Sakthivel, and Nagasuma Chandra. "Fast local alignment of protein pockets (FLAPP): a system-compiled program for large-scale binding site alignment." Journal of Chemical Information and Modeling 62.19 (2022): 4810-4819.
> > > > > >
> > > > > > # 3
> > > > > >
> > > > > > We sincerely thank the reviewer for their thoughtful feedback and for providing valuable references and examples. These contributions have greatly enriched our understanding and will help us refine our manuscript. We look forward to further engaging with the reviewer and deeply value the opportunity to learn from their expertise.

---

> > > > > > > ### Author Response · Authors · 2024-11-25
> > > > > > >
> > > > > > > Dear Reviewer,
> > > > > > >
> > > > > > > We have now obtained the delta scores using Glide XP:
> > > > > > >
> > > > > > > | Method            | Glide sp | Glide xp |
> > > > > > > |-------------------|----------|----------|
> > > > > > > | Reference ligand | 2.686    | 3.509    |
> > > > > > > | Pocket2Mol        | 0.531    | 0.553    |
> > > > > > > | TargetDiff        | 0.325    | 0.421    |
> > > > > > > | MolCRAFT          | 0.973    | 1.301    |
> > > > > > >
> > > > > > > As you suggested, switching to a more accurate docking approach indeed results in higher delta scores for both the reference ligands and the various methods. This confirms your expectations. Notably, the rankings of the different methods remain consistent and continue to fall significantly behind the reference ligand. Therefore, the conclusions and insights derived from the original results remain unchanged.
> > > > > > >
> > > > > > > We greatly appreciate your valuable suggestion, which has allowed us to present a more reliable and robust analysis. Thank you for helping us strengthen our findings.
> > > > > > >
> > > > > > > We would also like to confirm if our previous responses have satisfactorily addressed your concerns and questions. We have learned a lot during this rebuttal period, and we hope to continue this constructive discussion. This would provide us with further opportunities to clarify our work and gain valuable suggestions from you to strengthen it further.
> > > > > > >
> > > > > > > Best regards,
> > > > > > >
> > > > > > > Authors

---

> > > > > > > > ### Comment · Reviewer_pELf · 2024-11-25
> > > > > > > > **Response to authors**
> > > > > > > >
> > > > > > > > **I have raised my score to 6 to be above the acceptance**
> > > > > > > >
> > > > > > > > Thank you to the authors for an interesting discussion and the extra results and explanations. I know the discussion period is quite constrained. Overall, I agree that the proposed metrics will be valuable to better assess SBDD models (*except FDA similarity which we both agree on*). **I will reply to the specific points where I have suggestions on additional metrics/results that could improve the presentation (in my opinion).** For the points not replied to, I agree with the authors and thank you for the clarifications/further explanations.
> > > > > > > >
> > > > > > > > ### 2.2 Regarding Delta score
> > > > > > > >
> > > > > > > > Firstly, I have seen the new XP results the authors just posted. Thank you for running this experiment. I think these results should be included in your main text - maybe labeled Delta Score (SP) and Delta Score (XP). Regarding using Glide to enforce specific interactions, I agree this requires human labelling so it is hard to scale. However, every target comes from a paper which reports the discovery of the corresponding active. I know this is a lot of work, which is why I am suggesting the following as interesting future work: if all the important interactions are extracted and annotated and then Glide SP docking performed, I would suspect the Delta score will increase a lot.
> > > > > > > >
> > > > > > > > ### 2.5 Regarding the similarity metrics
> > > > > > > >
> > > > > > > > It would be interesting to report the enrichment/etc. results by partitioned active sets. This would answer the question: "For the models that generate molecules with similarity to known actives, are they more similar to more potent ones?"
> > > > > > > >
> > > > > > > > ### Other metrics
> > > > > > > >
> > > > > > > > Generally, it seems Pocket2Mol generates smaller ligands. As the authors have also discussed in the paper, molecule size has an effect on docking score. It should be just a little effort to additionally report molecule weight and/or ligand efficiency (docking score / # heavy atoms) for all the models in the tables. I think this information adds important context.

---

> > > > > > > > > ### Author Response · Authors · 2024-11-26
> > > > > > > > >
> > > > > > > > > We sincerely thank the reviewer for their thoughtful feedback and for recognizing the value of our work. We are delighted that the reviewer considers our proposed metrics valuable for improving the assessment of SBDD models and that our explanations have addressed their concerns. We also deeply appreciate the reviewer’s expertise in the SBDD domain, as reflected in their detailed perspectives and numerous references.
> > > > > > > > >
> > > > > > > > > Regarding Delta Score, we will include the Delta Score (XP) results in the main text. We fully agree that incorporating human labeling for target docking could enhance the accuracy of delta scores. Thank you for suggesting this as a promising avenue for future work.
> > > > > > > > >
> > > > > > > > > For the suggestion regarding similarity metrics, we agree that this is an excellent idea and will consider it as an important direction for future research.
> > > > > > > > >
> > > > > > > > > Regarding the addition of molecular weight efficiency, we believe this is an outstanding suggestion and will incorporate it into our paper.
> > > > > > > > >
> > > > > > > > > Once again, we sincerely thank the reviewer for their recognition of our work and for fostering a constructive and meaningful discussion with us.

---

### Official Review · Reviewer_hHxU · 2024-10-29

**Soundness:** 3
**Presentation:** 2
**Contribution:** 2
**Rating:** 8
**Confidence:** 3

**Summary:**

This paper introduces a three part framework to assess SBDD-generated molecules. The first part is similarity between generated molecules and known active compounds, achieved using molecular fingerprints. The second part is a virtual screening metric, a combination of BEDROC and EF, to measure the ability of generated molecules to distinguish between active and inactive compounds. The final part is binding affinity to a target, in which Vina and delta scores, and DrugCLIP were used to achieve this. Results show that amongst the used models, MolCRAFT dominated across multiple metrics.

**Strengths:**

- The problem that this paper attempts to address is one of high importance. Many generative/optimization models in this field rely on computational property predictors, which may not be accurate. The alternative would be to conduct real-world wet-lab experiments, which in many cases, is not feasible.
- The three proposed assessment techniques capture different aspects of the generated molecules and make for a combination that avoids redundancy.
- The datasets used (especially the testing set) seem to be satisfactory.

**Weaknesses:**

- The authors state: “This provides a practical alternative to testing deep learning-generated molecules directly in wet-lab experiments.” I believe this is quite a bold statement and, unfortunately, am not fully convinced of this. While the authors provide a neat framework and results, I think more experiments on larger datasets are required to show that this statement holds true. Moreover, it would be much more convincing if the authors are able to demonstrate correlation between their metrics and actual wet-lab results on a subset of compounds.


- I understand the rationale behind using similarity to known active compounds as an assessment. However, my concern is that this can be limiting – we can generate very desirable molecules that may be extremely different from the active compounds.



Minor edits:
- Line 118: generated by docking software instead of real complex → complexes
- Line 307: Actually, ligands in PDBbind has a higher docking score → have
- This paragraph needs to be better written: "We refined the PDBbind dataset by excluding complexes with nuclear attachment and inaccurately recorded ligands. Then split into a 9:1 training and validation set. To assess the SBDD model’s generalization across diverse pockets, we removed samples with similar pockets using FLAPP for pocket alignment and similarity assessment. We remove all pockets from the training set with align rate more than 0.6 or 0.9 to the test set pocket. After the removal, the 0.6 version has 12344 pairs remaining while the 0.9 version has 17519 pairs remaining."
- Line 485: We do some visualizations → present

- Figure 6: Be consistent with wording/capitalization:
    - Molcraft Generated Mol
    - Molecule generated by TargerDiff

- Figure 5 captions: include spaces after “(a) ...” and “(b) ...”

**Questions:**

- How would your method evaluate a generated molecule that has a very different structure from known actives, but demonstrates strong binding affinity in docking simulations?

- Can you please clarify why you are so confident that this is a practical alternative to wet-lab experiments?

---

> ### Author Response · Authors · 2024-11-19
>
> Thank you for taking the time to read our paper and for your thoughtful review. We greatly appreciate your recognition of the importance of the problem we are addressing and your acknowledgment of the value of our new perspective on evaluation metrics. We have corrected all the typos you mentioned in the manuscript. We have corrected some errors in lines 118, 307, 485, as well as in Figures 5 and 6, and we have provided a clearer description of the section you mentioned. Thank you for your careful reading. Next, we will provide detailed responses to your questions and concerns regarding our work.
>
> ## 1.Regarding the statement on alternative to wet-lab experiments.
>
> We sincerely apologize for any confusion caused by this claim. The original text stated, “This provides a practical alternative to testing deep learning-generated molecules directly in wet-lab experiments.” We acknowledge that this sentence may have been ambiguous.
>
> To clarify, **we did not intend to claim that our evaluation framework can replace wet-lab experiments.** What we intended to say is that since models like MolCRAFT demonstrate reasonable performance in virtual screening, then it is not mandatory to directly use the generated molecules in wet-lab experiments. **Alternatively, the generated molecules can serve as references for virtual screening, and the screened molecules can then be tested in wet-lab experiments.**
>
> We have updated this sentence in the paper to eliminate any ambiguity. We apologize again for any lack of clarity in the original text.
>
>
> ## 2.Regarding the limitation of similarity to known active compounds
>
> We appreciate the reviewer for raising this question, allowing us the opportunity to provide further clarification.
>
> The motivation behind proposing the similarity metric is that human experts modifying generated molecules is also a method of deploying SBDD models. Therefore, if a model’s output is closer to a truly active one, it becomes easier for human experts to modify these molecules into drug candidates.
>
> From a machine learning perspective, **the model does not have access to information about actives during inference and sampling.** Our rationale is that if the model can generate molecules similar to experimentally validated actives for a well-studied target, then **it should be able to generalize to new targets that have not been thoroughly investigated or lack known actives, producing molecules similar to the true ligands of these new targets.** The functional groups and scaffolds present in these generated molecules can assist human experts in identifying truly binding compounds. After all, there are still many targets that have not been sufficiently studied.
>
> Come back to your concern. Of course, there might be entirely new drug-like small molecules that do not resemble any known drugs. However, our similarity metric is not aimed at evaluating a single molecule but rather assesses a generative model by taking the maximum similarity across many generated molecules. Our reasoning is that **if a model consistently generates small molecules across various targets that are all far from the known actives, the model’s ability to generate actives may be questionable.**
>
> Finally, we acknowledge that during evaluation, if a generated molecule is highly desirable yet significantly different from known active compounds, our similarity-based metrics may not assign a high score to that molecule. However, whether such a molecule is genuinely effective remains uncertain without actual wet-lab experiments. From an evaluation standpoint, this does not invalidate our evaluation framework, as its purpose is to provide **a structured and fair assessment based on available data.**
>
> That said, we recognize that no metric is perfect. **Our similarity metric is not intended to replace theoretical estimation-based metrics like Vina docking scores or QED. Instead, it is designed to provide an additional perspective that supplements these metrics**.
>
> In our evaluation framework, we assess the effectiveness of SBDD models across **three distinct levels**. We introduce the Delta Score and DrugCLIP Score to complement traditional docking scores as theoretical estimation-based metrics. Additionally, we incorporate similarity to known actives as another evaluation dimension. **By combining these metrics, we aim to minimize the risk of drawing incorrect conclusions and provide a more comprehensive evaluation.**

---

> ### Author Response · Authors · 2024-11-19
>
> ## 3. Regarding the scenario where docking scores and similarity metrics do not align.
>
> We believe this also addresses your question regarding how our method would evaluate a generated molecule with a structure vastly different from known actives but demonstrating strong binding affinity in docking simulations. Our similarity-based and virtual screening-based metrics are not intended for molecule-level evaluation but rather serve as benchmarks for model-level assessment. **In other words, if an individual molecule receives a low similarity score to all known active compounds, we are not suggesting that the molecule should be dismissed outright. However, if a model predominantly generates such molecules, its overall capability would be called into question.**
>
> Come back to the question. If a molecule exhibits a significantly different structure from known actives yet demonstrates strong binding affinity in docking simulations, our framework allows for additional evaluation using the Delta Score and DrugCLIP score. We believe these metrics provide a more robust and reliable assessment of binding affinity at the molecule level compared to conventional docking scores.
>
>
> ## 4. Regarding the question of why we are so confident that this is a practical alternative to wet-lab experiments
>
> Similar to the previous response, we want to clarify that **we did not want to claiming our metrics can replace wet-lab experiments.** We apologize for any lack of clarity and have updated the paper to revise the sentence that may have caused this confusion.
>
> Our claim is that our metrics provide a better simulation of real-world experiments. For example, in the virtual screening metrics, **all active and inactive molecules used in benchmarks like DUD-E and LIT-PCBA are wet-lab validated**, meaning that better virtual screening performance directly correlates with higher wet-lab success rates. In contrast, traditional metrics like the Vina docking score are purely theoretical estimations based on specific formulas and are not directly indicative of wet-lab outcomes.
>
> Fortunately, some approaches using SBDD models for virtual screening have already been validated by wet-lab experiments. For example, the TamGen team [1] sought commercially available compounds similar to those generated by their model from a 446k compound library. They successfully identified 159 analogs, and five of these analogs demonstrated significant inhibitory effects in a wet-lab ClpP1P2 peptidase activity assay. Similarly, the Pocket Crafter project [2] employed SAR enrichment analysis to extract valuable scaffolds from the generated molecules, using these scaffolds to conduct ligand-similarity-based virtual screening. They done wet-lab experiment on the 2029 compounds searched from  Novartis internal archived diverse library and led to the finding of WM-662 and Compound 1 [3], the known WBM pocket binding molecules.We hope that our metrics can provide a broader perspective for evaluating the usefulness of SBDD models.
>
> [1] Wu, Kehan, et al. "TamGen: drug design with target-aware molecule generation through a chemical language model." Nature Communications 15.1 (2024): 9360.
>
> [2] Shen, Lingling, et al. "Pocket Crafter: a 3D generative modeling based workflow for the rapid generation of hit molecules in drug discovery." Journal of Cheminformatics 16.1 (2024): 33.
>
> [3] Ding, Jian, et al. "Discovery and structure-based design of inhibitors of the WD repeat-containing protein 5 (WDR5)–MYC interaction." Journal of Medicinal Chemistry 66.12 (2023): 8310-8323.

---

> > ### Comment · Reviewer_hHxU · 2024-11-22
> > **Response to Authors**
> >
> > Dear authors,
> >
> > Thank you for your thorough response. My rating will remain as "accept".

---

> > > ### Author Response · Authors · 2024-11-22
> > >
> > > We sincerely appreciate the time and effort you have devoted to reviewing our paper and providing thoughtful and constructive suggestions, which will greatly help us improve our work!

---

### Author Response · Authors · 2024-12-04
**Summary of Reviewer-Author Discussion Period**

Dear AC and all reviewers,

We sincerely appreciate your time and efforts in reviewing our paper. During the rebuttal period, the reviewers provided invaluable feedback that allowed us to refine our work and address any areas of potential ambiguity. We sincerely thank the reviewers for their engagement in multiple rounds of discussions, which provided us with valuable insights and learning opportunities. **We are particularly pleased that, following the discussions, all reviewers acknowledged the value of our proposed benchmark framework to the structure-based drug design community.**

Reviewers have recognized the following **merits of our work**:

- **We are tackling a significant problem. [hHxU, pELf, WYJC]**: Proposing new metrics in SBDD is a crucial step toward enabling a more comprehensive evaluation, as it reduces reliance on existing computational property predictors, which are often susceptible to exploitation.

- **Proposed metrics will be valuable/useful to better assess SBDD models. [WYJC, pELf, hHxU, 8aMH]** : We believe that beyond using SBDD-generated molecules directly as drug candidates, they can be applied in more practical ways, such as in ligand-based virtual screening. Our proposed metrics offer a more comprehensive evaluation of the usefulness and effectiveness of current SBDD models across the entire drug discovery pipeline.

- **Our test set, training set, and structure-based splitting method have been positively recognized.[hHxU, pELf]** ： These ensure that our testing also reflects the generalization and transferability of different models. We retrained all models using these datasets to maintain fairness and reliability as a benchmark.


During the rebuttal process, we thoroughly addressed the reviewers’ comments and suggestions by providing detailed explanations, evidence, and incorporating feedback to enhance the manuscript. We demonstrated **strong alignment** between virtual screening metrics and wet-lab performance through examples that highlighted the importance of evaluating SBDD models under diverse deployment strategies. We clarified the rationale for using similarity-based metrics, explaining that **as a benchmarking paper**, our approach **does not constrain chemical space exploration**. We added the test results of an **RL and genetic algorithm based model**, RGA, to enrich our study. Additionally, we emphasized the robustness of our structure-based dataset splitting strategy, which incorporates a **highly diverse and carefully curated test set**, referencing **prior wet-lab studies** and **simulations on underexplored targets** to demonstrate the framework’s applicability, mitigate overfitting risks, and ensure reliability on untested targets, **effectively addressing concerns related to transferability**.

We are confident that all questions and suggestions raised by the reviewers have been thoroughly addressed during the rebuttal period, and our contributions have been explicitly recognized by the reviewers.


Based on the reviewer’s suggestions, we made the following revisions to the manuscript:

- Added more detailed explanations of the reliability of virtual screening and similarity-based metrics, along with examples from real-world studies **[8aMH]**.
- Included results of delta score calculations using Glide XP **[pELf]**.
- Added the Atom Efficiency results for different models as a reference **[pELf]**.
- Moved the FDA similarity metric and results to the appendix, as it might be too strict **[pELf]**.
- Included visualizations of docking poses in the appendix **[WYJC]**.
- Added the RGA method based on genetic and reinforcement learning to the appendix **[WYJC]**.
- Provided a more detailed explanation of Table 1, including the test set and partitioning method **[WYJC]**.
- Fixed various spelling errors and typos **[hHxU]**.
- Fixed some sentences that may cause ambiguity, and make our motivation more clear.

In summary, the **main contribution** of our work lies in proposing a set of metrics to evaluate SBDD models from a broader and more reliable perspective and providing a rigorously partitioned training and test set for fair comparisons. We also conducted comprehensive testing and analysis of current leading 3D-SBDD models. We hope our metrics can fairly assess the applicability of models in the drug discovery pipeline and provide insights from metric comparisons to guide the design of better models in the future.

We sincerely thank the reviewers for their constructive feedback and thoughtful suggestions, which have significantly enhanced the quality of our paper. We will incorporate these insights into our future research to further contribute to advancements in the field of SBDD. Once again, we deeply appreciate your time and invaluable input.

Best regards,

The Authors

---

### Meta-Review · Area_Chair_hvAq · 2024-12-18

**Metareview:**

This paper proposes a framework for evaluating structure-based drug design (SBDD) based on practical metrics that may be better aligned with the efficacy of the designed molecules in real-world applications.
Reviewers generally agree that the proposed evaluation framework addresses an important challenge in SBDD and has the potential to improve the reliability of the evaluation results.
There have been moderate concerns regarding the novelty of the proposed approach, practical limitations on its expected impact in enhancing SBDD, and the need for clarifications.
The authors' rebuttals have addressed the concerns raised by the reviewers to some extent.

**Additional Comments On Reviewer Discussion:**

The authors and all reviewers have actively engaged in discussing the original concerns and doubts.
Their fruitful discussion and the authors' active response have been helpful in addressing many of these initial concerns, which has enhanced the confidence of the reviewers' overall evaluations.
There is a general agreement among the reviewers regarding the practical contribution of the current paper and there don't seem to be any major concerns that would need to be addressed.

---

### Decision · Program_Chairs · 2025-01-22

Accept (Poster)